# Graph Classification via Reference Distribution Learning: Theory and Practice

**Zixiao Wang**     **Jicong Fan**[*]
School of Data Science
The Chinese University of Hong Kong, Shenzhen
zixiaowang@link.cuhk.edu.cn     fanjicong@cuhk.edu.cn

## Abstract

Graph classification is a challenging problem owing to the difficulty in quantifying the similarity between graphs or representing graphs as vectors, though there have been a few methods using graph kernels or graph neural networks (GNNs). Graph kernels often suffer from computational costs and manual feature engineering, while GNNs commonly utilize global pooling operations, risking the loss of structural or semantic information. This work introduces Graph Reference Distribution Learning (GRDL), an efficient and accurate graph classification method. GRDL treats each graph's latent node embeddings given by GNN layers as a discrete distribution, enabling direct classification without global pooling, based on maximum mean discrepancy to adaptively learned reference distributions. To fully understand this new model (the existing theories do not apply) and guide its configuration (e.g., network architecture, references' sizes, number, and regularization) for practical use, we derive generalization error bounds for GRDL and verify them numerically. More importantly, our theoretical and numerical results both show that GRDL has a stronger generalization ability than GNNs with global pooling operations. Experiments on moderate-scale and large-scale graph datasets show the superiority of GRDL over the state-of-the-art, emphasizing its remarkable efficiency, being at least 10 times faster than leading competitors in both training and inference stages. The source code of GRDL is available at https://github.com/jicongfan/GRDL-Graph-Classification.

## 1 Introduction

Graphs serve as versatile models across diverse domains, such as social networks [Wang et al., 2018], biological compounds [Jumper et al., 2021], and the brain [Ktena et al., 2017]. There has been considerable interest in developing learning algorithms for graphs, such as graph kernels [Gärtner et al., 2003, Shervashidze et al., 2011, Chen et al., 2022b] and graph neural networks (GNNs) [Kipf and Welling, 2016, Defferrard et al., 2016, Gilmer et al., 2017]. GNNs have emerged as powerful tools, showcasing state-of-the-art performance in various graph prediction tasks [Veličković et al., 2017, Gilmer et al., 2017, Hamilton et al., 2017, Xu et al., 2018, Sun et al., 2019, You et al., 2021, Ying et al., 2021, Liu et al., 2022b, Chen et al., 2022a, Xiao et al., 2022, Sun et al., 2023, Sun and Fan, 2024, Sun et al., 2024]. Despite the evident success of GNNs in numerous graph-related applications, their potential remains underutilized, particularly in the domain of graph-level classification.

Current GNNs designed for graph classification commonly consist of two components: the embedding of node features through message passing [Gilmer et al., 2017] and subsequent aggregation by some permutation invariant global pooling (also called readout) operations [Xu et al., 2018]. The primary purpose of pooling is to transform a graph's node embeddings, a matrix, into a single vector.

---

[*]Corresponding author

Empirically, pooling operations play a crucial role in classification [Ying et al., 2018]. However, these pooling operations tend to be naive, often employing methods such as simple summation or averaging. These functions collect only first-order statistics, leading to a loss of structural or semantic information. In addition to the conventional sum or average pooling, more sophisticated pooling operations have shown improvements in graph classification [Li et al., 2015, Ying et al., 2018, Lee et al., 2019, 2021, Buterez et al., 2022, Yu et al., 2024], but they still carry the inherent risk of information loss.

Different from graph kernel methods and existing GNN methods, we propose a novel GNN method that classifies the nodes' embeddings themselves directly, thus avoiding the global pooling step. In our method, we treat the nodes' latent representations of each graph, learned by a neural network, as a discrete distribution and classify these distributions into $K$ different classes. The classification is conducted via measuring the similarity between the latent graph's distributions and $K$ discriminative reference discrete distributions. The reference distributions can be understood as nodes' embeddings of representative virtual graphs from $K$ different classes, and they are jointly learned with the parameters of the neural network in an end-to-end manner. To evaluate our method, we analyze the generalization ability of our model both theoretically and empirically. Our contributions are two-fold.

- We propose a novel graph classification method GRDL that is efficient and accurate.
  - GRDL does not require any global pooling operation and hence effectively preserves the information of node embeddings.
  - Besides its high classification accuracy, GRDL is scalable to large graph datasets and is at least ten times faster than leading competitors in both training and inference stages.
- We provide theoretical guarantees, e.g. generalization error bounds, for GRDL.
  - The result offers valuable insights into how the model performance scales with the properties of graphs, neural network structure, and reference distributions, guiding the model design.
  - For instance, the generalization bounds reveal that the references' norms and numbers have tiny impacts on the generalization, which is also verified by the experiments.
  - More importantly, we theoretically prove that GRDL has a stronger generalization ability than GNNs with global pooling operations.

The rest of this paper is organized as follows. We introduce our model in Section 2 and analyze the generalization ability in Section 3. Related works are discussed in Section 4. Section 5 presents the numerical results on 11 benchmarks in comparison to 12 competitors.

## 2 Proposed Approach

### 2.1 Model Framework

Following convention, we denote a graph with index $i$ by $G_i = (V_i, E_i)$, where $V_i$ and $E_i$ are the vertex (node) set and edge set respectively. Given a graph dataset $\mathcal{G} = \{(G_1, y_1), (G_2, y_2), \ldots, (G_N, y_N)\}$, where $y_i \in \{1, 2, \ldots, K\}$ is the associated label of $G_i$ and $y_i = k$ means $G_i$ belongs to class $k$, the goal is to learn a classifier $f$ from $\mathcal{G}$ that generalizes well to unseen graphs. Since in many scenarios, each node of a graph has a feature vector $\mathbf{x}$ and the graph is often represented by an adjacency matrix $\mathbf{A}$, we also write $G_i = (\mathbf{A}_i, \mathbf{X}_i)$ for convenience, where $\mathbf{A}_i \in \mathbb{R}^{n_i \times n_i}$, $\mathbf{X}_i \in \mathbb{R}^{n_i \times d_0}$, $n_i = |V_i|$ is the number of nodes of graph $i$, and $d_0$ denotes the number of features. We may alternatively denote the graph dataset as $\mathcal{G} = \{((\mathbf{A}_1, \mathbf{X}_1), y_1), ((\mathbf{A}_2, \mathbf{X}_2), y_2), \ldots, ((\mathbf{A}_N, \mathbf{X}_N), y_N)\}$.

Our approach is illustrated in Figure 1. For graph classification, we first use a GNN, denoted as $f_G$, to transform each graph to a node embedding matrix $\mathbf{H}_i \in \mathbb{R}^{n_i \times d}$ that encodes its properties, i.e.,

$$\mathbf{H}_i = f_G(G_i) = f_G(\mathbf{A}_i, \mathbf{X}_i), \tag{1}$$

where $f_G \in \mathcal{F}_G$ and $\mathcal{F}_G$ denotes a hypothesis space. The remaining task is to classify $\mathbf{H}_i$ without global pooling. Direct classification of node embeddings is difficult due to two reasons:

(i) Different graphs have different numbers of nodes, i.e. in general, $n_i \neq n_j$ if $i \neq j$.
(ii) The node embeddings of each graph are permutation invariant, namely, $\mathbf{PH}_i$ and $\mathbf{H}_i$ represent the same graph for any permutation matrix $\mathbf{P}$.

However, the two properties are naturally satisfied if we treat the node embeddings of each graph as a discrete distribution. Specifically, each $\mathbf{H}_i$ is a discrete distribution and each row of $\mathbf{H}_i$ is an

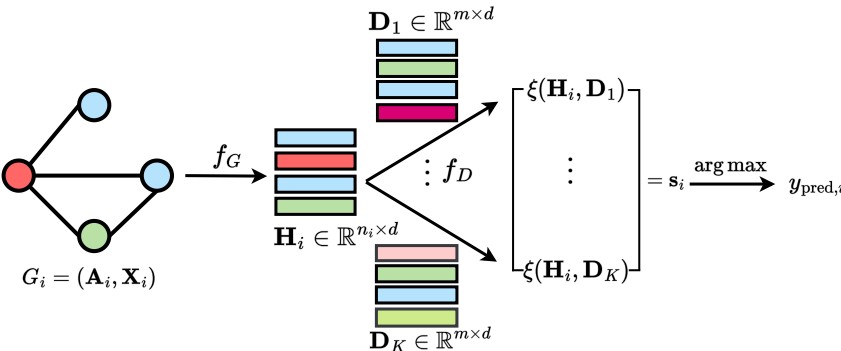

Figure 1: The GRDL framework. Classification involves using a GNN $f_G$ to encode a graph's information into a node embedding distribution. The similarities between the node embeddings and $K$ reference distributions are calculated by the reference module $f_D$. The graph is assigned the label of the reference that exhibits the highest similarity.

outcome of the distribution. There is no order between the outcomes in each distribution. Also, different distributions may have different numbers of outcomes. Before introducing our method in detail, we first give a toy example where the commonly used mean and max pooling operations fail.

**Example 2.1.** *Suppose two graphs $G_1$ and $G_2$ have self-looped adjacency matrices $\tilde{\mathbf{A}}_1 = [1\ 1\ 1\ 0; 1\ 1\ 0\ 1; 1\ 0\ 1\ 1; 0\ 1\ 1\ 1]$ and $\tilde{\mathbf{A}}_2 = [1\ 0\ 0\ 0; 0\ 1\ 1\ 0; 0\ 1\ 1\ 0; 0\ 0\ 0\ 1]$ respectively, and have one-dimensional node features $\mathbf{X}_1 = [3\ 6\ 9\ 12]^\top$ and $\mathbf{X}_2 = [6\ 6\ 9\ 9]^\top$ respectively. Let $\hat{\mathbf{A}}_i$ be the normalized adjacency matrices, i.e., $\hat{\mathbf{A}}_i = diag(\tilde{\mathbf{A}}_i \mathbf{1})^{-1/2} \tilde{\mathbf{A}}_i diag(\tilde{\mathbf{A}}_i \mathbf{1})^{-1/2}$. Performing the neighbor aggregation $\mathbf{H}_i = \hat{\mathbf{A}}_i \mathbf{X}_i$, $i = 1, 2$, we obtain $\mathbf{H}_1 = [6\ 7\ 8\ 9]^\top$ and $\mathbf{H}_2 = [6\ 7.5\ 7.5\ 9]^\top$. We see that $\mathrm{mean}(\mathbf{H}_1) = \mathrm{mean}(\mathbf{H}_2) = 7.5$ and $\max(\mathbf{H}_1) = \max(\mathbf{H}_2) = 9$. This means the simple mean and max pooling operations failed to distinguish the two graphs. In contrast, our method treats $\mathbf{H}_1$ and $\mathbf{H}_2$ as two different discrete distributions and hence is able to distinguish the two graphs. Note that incorporating a learnable parameter $\mathbf{W}$, i.e., $\mathbf{H}_i = \hat{\mathbf{A}}_i \mathbf{X}_i \mathbf{W}$, or performing multiple times of neighbor aggregation does not change the conclusion.*

We propose to classify the discrete distributions $\{\mathbf{H}_1, \mathbf{H}_2, \ldots, \mathbf{H}_N\} \triangleq \mathcal{H}$ by a reference layer $f_D$. The classification involves measuring the similarity between $\mathbf{H}_i$ and $K$ reference discrete distributions $\{\mathbf{D}_1, \mathbf{D}_2, \ldots, \mathbf{D}_K\} \triangleq \mathcal{D}$ that are discriminative. Each $\mathbf{D}_k \in \mathbb{R}^{m_k \times d}$ can be understood as node embeddings of a virtual graph from the $k$-th class, $k \in [K]$. We make $m_1 = \cdots = m_K = m$ for convenience. Letting $\xi$ be a similarity measure between two discrete distributions, then

$$s_{ik} := \xi(\mathbf{H}_i, \mathbf{D}_k), \quad i \in [N], \ k \in [K]. \tag{2}$$

This forms a matrix $\mathbf{S} = [\mathbf{s}_1, \mathbf{s}_2, \ldots, \mathbf{s}_N]^\top$ where

$$\mathbf{s}_i = f_D(\mathbf{H}_i) = [s_{i1}, s_{i2}, \ldots, s_{iK}]^\top \in \mathbb{R}^K, \tag{3}$$

$f_D \in \mathcal{F}_D$ and $\mathcal{F}_D$ denotes a hypothesis space induced by the reference layer. References in $\mathcal{D}$ are parameters of the reference layer and are jointly learned with node-embedding network parameters in an end-to-end manner. Now combining Equation (1) and Equation (3), we arrive at

$$\mathbf{s}_i = f_D(f_G(G_i)), \quad i \in [N]. \tag{4}$$

$f_D \circ f_G$ calculates $\mathbf{s}_i$, representing similarities between $G_i$ and all references. We get $G_i$'s label by

$$y_{\mathrm{pred},i} = \arg\max_k s_{ik}. \tag{5}$$

To train the model, we first use the softmax function to convert $\mathbf{s}_i$ to a label vector $\hat{\mathbf{y}}_i = [\hat{y}_{i1}, \ldots, \hat{y}_{iK}]^\top$, where

$$\hat{y}_{ik} = \frac{\exp(s_{ik})}{\sum_{j=1}^K \exp(s_{ij})}, \quad k \in [K]. \tag{6}$$

Using the cross-entropy loss, we minimize

$$\mathcal{L}_{\text{CE}} = -\frac{1}{N} \sum_{i=1}^{N} \sum_{k=1}^{K} y_{ik} \log \hat{y}_{ik}. \tag{7}$$

Intuitively, the reference distributions in $\mathcal{D}$ should be different from each other to ensure discriminativeness. Therefore, we also consider the following discrimination loss:

$$\mathcal{L}_{\text{Dis}} = \sum_{k} \sum_{k' \neq k} \xi(\mathbf{D}_k, \mathbf{D}_{k'}). \tag{8}$$

Then we solve the following problem:

$$\min_{f_G \in \mathcal{F}_G, f_D \in \mathcal{F}_D} \mathcal{L}_{\text{CE}} + \lambda \mathcal{L}_{\text{Dis}}, \tag{9}$$

where $\lambda \geq 0$ is a hyperparameter. We call (9) Graph Classification via Reference Distribution Learning (GRDL). Specific designs of $\mathcal{F}_G$ and $\mathcal{F}_D$ are detained in the next section.

## 2.2 Design of $\mathcal{F}_G$ and $\mathcal{F}_D$

We get GRDL's network $\mathcal{F}$ by concatenating the node embedding module and the reference module:

$$\mathcal{F} := \mathcal{F}_D \circ \mathcal{F}_G. \tag{10}$$

**Design of $\mathcal{F}_G$** We use an $L$-layer message passing network as our node embedding module $\mathcal{F}_G$:

$$\mathcal{F}_G := \mathcal{F}^L \circ \mathcal{F}^{L-1} \circ \cdots \circ \mathcal{F}^1. \tag{11}$$

$\mathcal{F}^l$ is the $l$-th message passing layer (e.g. a GIN layer [Xu et al., 2018]) that updates the representation of a node by aggregating representations of its neighbors, meaning

$$a_v^{(l)} = \text{AGGREGATE}^{(l)} \left( \left\{ h_u^{(l-1)} : u \in \mathcal{N}(v) \right\} \right), \quad h_v^{(l)} = \text{COMBINE}^{(l)} \left( h_v^{(l-1)}, a_v^{(l)} \right) \tag{12}$$

where $h_v^{(l)}$ is the feature vector of node $v$ produced by the $l$-th layer $\mathcal{F}^l$. Different GNNs have different choices of $\text{COMBINE}^{(l)}(\cdot)$ and $\text{AGGREGATE}^{(l)}(\cdot)$.

**Design of $\mathcal{F}_D$** Based on (3), the hypothesis space defined by the reference layer is

$$\mathcal{F}_D := \{ \mathbf{H}_i \mapsto \mathbf{s}_i \in \mathbb{R}^K : s_{ik} = \xi(\mathbf{H}_i, \mathbf{D}_k), \mathbf{D}_k \in \mathbb{R}^{m \times d} \}. \tag{13}$$

In our work, we choose $\xi(\cdot, \cdot)$ to be the negative squared Maximum Mean Discrepancy (MMD). Initially used for two-sample tests, MMD is now widely used to measure the dissimilarity between distributions [Gretton et al., 2012a]. For an embedding $\mathbf{H} \in \mathbb{R}^{n \times d}$ and a reference $\mathbf{D} \in \mathbb{R}^{m \times d}$,

$$\xi(\mathbf{H}, \mathbf{D}) = - \text{MMD}^2 (\mathbf{H}, \mathbf{D}) = - \left\| \frac{1}{n} \sum_{i=1}^{n} \phi(\mathbf{h}_i) - \frac{1}{m} \sum_{j=1}^{m} \phi(\mathbf{d}_j) \right\|_2^2$$

$$= \frac{2}{mn} \sum_{i=1}^{n} \sum_{j=1}^{m} \phi(\mathbf{h}_i)^\top \phi(\mathbf{d}_j) - \frac{1}{n^2} \sum_{i=1}^{n} \sum_{i'=1}^{n} \phi(\mathbf{h}_i)^\top \phi(\mathbf{h}_{i'}) - \frac{1}{m^2} \sum_{j=1}^{m} \sum_{j'=1}^{m} \phi(\mathbf{d}_j)^\top \phi(\mathbf{d}_{j'}) \tag{14}$$

where $\phi$ is some feature map, $\mathbf{h}_i^\top$ is the $i$-th row of $\mathbf{H}$, and $\mathbf{d}_j^\top$ is the $j$-th row of $\mathbf{D}$. The MMD in (14) is known as biased MMD [Gretton et al., 2012a] and its performance is almost the same as the unbiased one, in our experiments. Therefore we only present (14) here. Using kernel trick $k(\mathbf{x}, \mathbf{x}') = \phi(\mathbf{x})^\top \phi(\mathbf{x}')$, we obtain from (14) that

$$\xi(\mathbf{H}, \mathbf{D}) = \frac{2}{mn} \sum_{i=1}^{n} \sum_{j=1}^{m} k(\mathbf{h}_i, \mathbf{d}_j) - \frac{1}{n^2} \sum_{i=1}^{n} \sum_{i'=1}^{n} k(\mathbf{h}_i, \mathbf{h}_{i'}) - \frac{1}{m^2} \sum_{j=1}^{m} \sum_{j'=1}^{m} k(\mathbf{d}_j, \mathbf{d}_{j'}).$$

In this work, we employ the Gaussian kernel, i.e.,

$$k(\mathbf{x}, \mathbf{x}') = \exp \left( -\theta \| \mathbf{x} - \mathbf{x}' \|_2^2 \right) \tag{15}$$

where $\theta > 0$ is a hyperparameter. The Gaussian kernel defines an infinite-order polynomial feature map $\phi$, covering all orders of statistics of the input variable. Consequently, MMD with the Gaussian kernel characterizes the difference between two distributions across all moments. Actually, we found that, in GRDL, the Gaussian kernel often outperformed other kernels such as the polynomial kernel.

Several other statistical distances are available for measuring the difference between distributions, including Wasserstein distance and Sinkhorn divergence [Peyré and Cuturi, 2020]. However, their computational complexity is prohibitively high, making the model impractical for large-scale graph datasets. We also find, through experiments, that in our method, the classification performance of MMD is better than that of Wasserstein distance and Sinkhorn divergence as shown later in Table 1. These explain why we prefer MMD.

### 2.3 Algorithm Implementation

The $\theta$ in the Gaussian kernel (15) plays a crucial role in determining the statistical efficiency of MMD. Optimally setting of $\theta$ remains an open problem and many heuristics are available [Gretton et al., 2012b]. To simplify the process, we make $\theta$ learnable in our GRDL and rewrite $\xi$ as $\xi_\theta$. Our empirical results in Appendix D.5 show that GRDL with learnable $\theta$ performs better. For convenience, we denote all the parameters of $f_G$ as $\mathbf{w}$ and let $f_{\mathbf{w},\mathcal{D},\theta} = f_D \circ f_G$. Then we rewrite problem (9) as

$$\min_{\mathbf{w},\mathcal{D},\theta} -\frac{1}{N} \sum_{i=1}^{N} \sum_{k=1}^{K} y_{ik} \log \frac{\exp\left(f_{\mathbf{w},\mathcal{D},\theta}(G_i)_k\right)}{\sum_{j=1}^{K} \exp\left(f_{\mathbf{w},\mathcal{D},\theta}(G_i)_j\right)} + \lambda \sum_{k' \neq k} \xi_\theta(\mathbf{D}_k, \mathbf{D}_{k'}). \tag{16}$$

The (mini-batch) training of GRDL model is detailed in Algorithm 1 (see Appendix C).

## 3 Theoretical Analysis

In this section, we provide theoretical guarantees for GRDL, due to the following motivations:

- As the proposed approach is novel, it is necessary to understand it thoroughly using theoretical analysis, e.g., understand the influences of data and model properties on the classification.
- It is also necessary to provide guidance for the model design to guarantee high accuracy in inference stages.

### 3.1 Preliminaries

**Matrix constructions** We construct big matrices $\mathbf{X}$, $\mathbf{A}$ and $\mathbf{D}$, where $\mathbf{X} = \left[\mathbf{X}_1^\top, \mathbf{X}_2^\top, \ldots, \mathbf{X}_N^\top\right]^\top \in \mathbb{R}^{(\sum_i n_i) \times d}$; $\mathbf{A} = \mathrm{diag}(\mathbf{A}_1, \mathbf{A}_2, \ldots, \mathbf{A}_N) \in \mathbb{R}^{(\sum_i n_i) \times (\sum_i n_i)}$ is a block diagonal matrix, $\mathbf{D} = \left[\mathbf{D}_1^\top, \mathbf{D}_2^\top, \ldots, \mathbf{D}_K^\top\right]^\top \in \mathbb{R}^{Km \times d}$. The adjacency matrix with self-connectivity is $\tilde{\mathbf{A}} = \mathbf{A} + \mathbf{I}$. The huge constructed graph is denoted by $\mathbf{G} = (\tilde{\mathbf{A}}, \mathbf{X})$. This construction allows us to treat all graphs in dataset $\mathcal{G}$ as a whole and it is crucial for our derivation.

**Neural network** Previously, for a deterministic network $f \in \mathcal{F}$, its output after feeding forward a single graph is $f(G_i)$. However, we mainly deal with the huge constructed graph $\mathbf{G}$ in this section, and notation will be overloaded to $f(\mathbf{G}) = \mathbf{S} \in \mathbb{R}^{N \times K}$, a matrix whose $i$-th row is $f(G_i)^\top$.

We instantiate the message passing network as Graph Isomorphism Network (GIN) [Xu et al., 2018]. We choose to focus on GIN for two reasons. Firstly, the analysis on GIN is currently limited, most of the current bounds for GNNs don't apply for GIN [Garg et al., 2020, Liao et al., 2021, Tang and Liu, 2023]. The other reason is that GIN is used as the message-passing network in our numerical experiments. Notably, our proof can be easily adapted to other message-passing GNNs (e.g. GCN [Kipf and Welling, 2016]). GIN updates node representations as

$$h_v^{(l)} = \mathrm{MLP}^{(l)}\left((1 + \varepsilon^{(l)})h_v^{(l-1)} + \sum_{u \in \mathcal{N}(v)} h_u^{(l-1)}\right) \tag{17}$$

where $h_v^{(l)}$ denotes the node features generated by $l$-th GIN message passing layer. Let $\varepsilon^{(l)} = 0$ for all layers and suppose all MLPs have $r$ layers, the node updates can be written in matrix form as

$$\mathbf{H}^{(l)} = \sigma\left(\cdots \sigma\left(\left(\tilde{\mathbf{A}}\mathbf{H}^{(l-1)}\right)\mathbf{W}_1^{(l)}\right) \cdots \mathbf{W}_{r-1}^{(l)}\right)\mathbf{W}_r^{(l)} \tag{18}$$

where $\mathbf{W}_i^{(l)} \in \mathbb{R}^{d_{i-1}^{(l)} \times d_i^{(l)}}$ is the weight matrix, and $\mathbf{H}^{(l)}$ is the matrix of node features with $\mathbf{H}^{(0)} = \mathbf{X}$. $\sigma(\cdot)$ is the non-linear activation function. Let $\mathcal{F}^l$ be the function space induced by the $l$-th message passing layer, meaning

$$\mathcal{F}^l = \{(\tilde{\mathbf{A}}, \mathbf{H}^{(l-1)}) \mapsto \mathbf{H}^{(l)} : \mathbf{W}_i^{(l)} \in \mathcal{B}_i^{(l)}, i \in [r]\} \tag{19}$$

where $\mathcal{B}_i^{(l)}$ is some constraint set on the weight matrix $\mathbf{W}_i^{(l)}$ and $\mathbf{H}^{(l)}$ is given by (18). The $L$-layer GIN function space $\mathcal{F}_G$ is the composition of $\mathcal{F}^l$ for $l \in [L]$, i.e.,

$$\mathcal{F}_G = \mathcal{F}^L \circ \mathcal{F}^{L-1} \circ \cdots \circ \mathcal{F}^1 = \{\mathbf{G} \mapsto f^L(\cdots f^1(\mathbf{G})) : f^i \in \mathcal{F}^i, \forall i \in [L]\}. \tag{20}$$

Letting $s_{ik} = -\mathrm{MMD}^2(\mathbf{H}_i^{(L)}, \mathbf{D}_k)$, the reference layer defines the following function space

$$\mathcal{F}_D = \{\mathbf{H}^{(L)} \mapsto \mathbf{S} \in \mathbb{R}^{N \times K} : \mathbf{D}_k \in \mathbb{R}^{m \times d}, k \in [K]\}. \tag{21}$$

Our proposed network (GRDL) is essentially $\mathcal{F} := \mathcal{F}_D \circ \mathcal{F}_G$.

**Loss Function** Instead of the cross entropy loss (7), we consider a general loss function $l_\gamma(\cdot, \cdot)$ satisfying $0 \leq l_\gamma \leq \gamma$ to quantify the model performance. Importantly, this loss function is not restricted to the training loss because our generalization bound is optimization-independent. For instance, the loss function can be the ramp loss that is commonly used for classification tasks [Bartlett et al., 2017, Mohri et al., 2018]. Given a neural network $f \in \mathcal{F}$, we want to upper bound the model population risk of graphs and labels from an unknown distribution $\mathcal{X} \times \mathcal{Y}$

$$L_\gamma(f) := \mathop{\mathbb{E}}_{(G,y) \sim \mathcal{X} \times \mathcal{Y}} [l_\gamma(f(G), y))]. \tag{22}$$

Given the observed graph dataset $\mathcal{G}$ sampled from $\mathcal{X} \times \mathcal{Y}$, the empirical risk is

$$\hat{L}_\gamma(f) := \frac{1}{N} \sum_{i=1}^{N} l_\gamma(f(G_i), y_i), \tag{23}$$

of which (7) is just a special case. Appendix E provides more details about the setup and our idea.

## 3.2 Main Results

For convenience, similar to [Bartlett et al., 2017, Ju et al., 2023], we make the following assumptions.

**Assumption 3.1.** The following conditions hold for $\mathcal{F}_\gamma := \{(G, y) \mapsto l_\gamma(f(G), y) : f \in \mathcal{F}\}$:

(i) The activation function $\sigma(\cdot)$ is 1-Lipschitz (e.g. Sigmoid, ReLU).
(ii) The weight matrices satisfy $\mathbf{W}_i^{(l)} \in \mathcal{B}_i^{(l)} := \{\mathbf{W}_i^{(l)} : \|\mathbf{W}_i^{(l)}\|_\sigma \leq \kappa_i^{(l)}, \|\mathbf{W}_i^{(l)}\|_{2,1} \leq b_i^{(l)}\}$.
(iii) The constructed reference matrix satisfy $\|\mathbf{D}\|_2 \leq b_D$.
(iv) The Gaussian kernel parameter $\theta$ is fixed.
(v) The loss function $l_\gamma(\cdot, y) : \mathbb{R}^K \to \mathbb{R}$ is $\mu$-Lipschitz w.r.t $\|\cdot\|_2$ and $0 \leq l_\gamma \leq \gamma$.

**Theorem 3.2** (Generalization bound of GRDL). *Let $n = \min_i n_i$, $c = \|\tilde{\mathbf{A}}\|_\sigma$, and $\bar{d} = \max_{i,l} d_i^{(l)}$. Denote $R_G := c^{2L} \|\mathbf{X}\|_2^2 \ln(2\bar{d}^2) \big( \prod_{l=1}^{L} (\prod_{i=1}^{r} \kappa_i^{(l)})^2 \big) \big( \sum_{l=1}^{L} \sum_{i=1}^{r} \big( \frac{b_i^{(l)}}{\kappa_i^{(l)}} \big)^{2/3} \big)^3$. For graphs $\mathcal{G} = \{(G_i, y_i)\}_{i=1}^{N}$ drawn i.i.d from any probability distribution over $\mathcal{X} \times \{1, \ldots, K\}$ and references $\{\mathbf{D}_k\}_{k=1}^{K}, \mathbf{D}_k \in \mathbb{R}^{m \times d}$, with probability at least $1 - \delta$, every loss function $l_\gamma$ and network $f \in \mathcal{F}$ under Assumption 3.1 satisfy*

$$L_\gamma(f) \leq \hat{L}_\gamma(f) + 3\gamma \sqrt{\frac{\ln(2/\delta)}{2N}} + \frac{8\gamma + 24\sqrt{v_1 + v_2} \ln N + 24\gamma \sqrt{N v_2 \ln v_3}}{N}$$

*where $v_1 = \frac{64\theta K R_G \mu^2}{n}$, $v_2 = Km\bar{d}$, and $v_3 = \frac{24\sqrt{\theta N} b_D \mu}{\sqrt{m}}$.*

The bound shows how the properties of the neural network, graphs, reference distributions, etc, influence the gap between training error and testing error. A detailed discussion will be presented in Section 3.3. Some interesting corollaries of Theorem 3.2, e.g., misclassification rate bound, can be found in Appendix F.7. Besides small generalization error $L_\gamma(f) - \hat{L}_\gamma(f)$, a good model should have small empirical risk $\hat{L}_\gamma(f)$. The empirical risk $\hat{L}_\gamma(f)$ is typically a surrogate loss of misclassification

rate of training data and a lower misclassification rate implies a smaller $\hat{L}_\gamma(f)$. We now provide a guarantee for the correct classification of training data, namely small $\hat{L}_\gamma(f)$.

Notably, the node embeddings $\mathbf{H}_i$ from the $k$-th class as well as the reference distributions $\mathbf{D}_k$ are essentially some *finite samples from an underlying continuous distribution* $\mathbb{P}_k$. One potential risk is that, although the continuous distributions $\mathbb{P}_1, \mathbb{P}_2, \ldots, \mathbb{P}_K$ are distinct, we can only observe their finite samples and may fail to distinguish them from each other with MMD. Specifically, suppose a node embedding $\mathbf{H}_i$ is from the $k$-th class, although $0 = \mathrm{MMD}(\mathbb{P}_k, \mathbb{P}_k) < \mathrm{MMD}(\mathbb{P}_k, \mathbb{P}_j)$ for any $j \neq k$, it is likely that $\mathrm{MMD}(\mathbf{H}_i, \mathbf{D}_k) > \mathrm{MMD}(\mathbf{H}_i, \mathbf{D}_j)$ for some $j \neq k$. The following theorem provides the correctness guarantee for the training dataset $\mathcal{G}$:

**Theorem 3.3.** *All graphs in the training set $\mathcal{G}$ are classified correctly with probability at least $1 - \delta$ if*

$$\min_{i \neq j} \mathrm{MMD}(\mathbb{P}_i, \mathbb{P}_j) > \left( \tfrac{1}{\sqrt{m}} + \tfrac{1}{\sqrt{n}} \right) \left( 4 + 4\sqrt{\log \tfrac{2N}{\delta}} \right).$$

Theorem 3.3 implies that a larger reference distribution size $m$ benefits the classification accuracy of training data, resulting in a lower $\hat{L}_\gamma(f)$. Moreover, a larger $\min_{i \neq j} \mathrm{MMD}(\mathbb{P}_i, \mathbb{P}_j)$ also makes correct classification easier according to the theorem, justifying our usage of discriminative loss (8).

### 3.3 Bound Discussion and Numerical Verification

Let $\bar{\kappa} = \max_{i,l} \kappa_i^{(l)}$, $\bar{b} = \max_{i,l} \frac{b_i^{(l)}}{\kappa_i^{(l)}}$ and suppose $\delta$ is large enough, we simplify Theorem 3.2 as

$$L_\gamma(f) \leq \hat{L}_\gamma(f) + \tilde{\mathcal{O}}\big( \tfrac{\sqrt{v_1} + \gamma\sqrt{Nv_2}}{N} \big) \leq \hat{L}_\gamma(f) + \tilde{\mathcal{O}}\big( \tfrac{\mu\bar{b}\|\mathbf{X}\|_2 c^L (Lr)^{\frac{3}{2}} \bar{\kappa}^{Lr} \sqrt{\theta K/n}}{N} + \gamma\sqrt{\tfrac{Kmd}{N}} \big)$$

**I. Dependence on graph property** One distinctive feature of our bound is its dependence on the spectral norm of graphs' adjacency matrix. The large adjacency matrix $\tilde{\mathbf{A}}$ is a block-diagonal matrix, so its spectral norm $c = \|\tilde{\mathbf{A}}\|_\sigma = \max_{i \in [N]} \|\tilde{\mathbf{A}}_i\|_\sigma$. By Lemma F.8, incorporating $c^L$ is sufficient for any $L$-step GIN message passing. This result aligns with Ju et al. [2023], who achieved this conclusion via PAC-Bayesian analysis. Our derivation, based on the Rademacher complexity, provides an alternative perspective supporting this result. Notably, Liao et al. [2021] and Garg et al. [2020] proposed bounds scaling with graphs' maximum node degree, which is larger than the spectral norm of the graphs' adjacency matrix (Lemma F.18). Consequently, our bound is tighter.

**II. Use moderate-size message passing GIN** The bound scales with the size of the message passing GIN, following $\tilde{O}(c^L (Lr)^{\frac{3}{2}} \bar{\kappa}^{Lr})$. Empirical observations reveal $\bar{\kappa} > 1$, and we prove that $c > 1$ (refer to Lemma F.20). Therefore, when the message-passing GNN has sufficient expressive power (resulting in a small $\hat{L}_\gamma(f)$), a network with a smaller $L$ and $r$ may guarantee a tighter bound on the population risk compared to a larger one. Therefore, a promising strategy is to use a moderate-size message passing GNN. This is empirically supported by Figure 5 of Appendix D.7.

**III. Use moderate-size references** The bound scales with the size of reference distributions $m$ as $\tilde{O}(\sqrt{m})$. When $m$ is smaller, the bound tends to be tighter. However, if $m$ is too small, the model's expressive capacity is limited, potentially resulting in a large empirical risk $\hat{L}_\gamma(f)$, and consequently, a large population risk. Therefore, using moderate-size references is a promising choice, as supported by our empirical validation results in Appendix D.3 (see Figure 6).

**IV. Regularization on references norm barely helps** Regularizing the norm of references $\|\mathbf{D}\|_2$, i.e., reducing $b_D$, might be considered to enhance the model's generalization. However, it is important to note that $b_D$ only influences the term $v_3$ (in logarithm) in Theorem 3.2 and has a tiny influence on the overall bound. Conversely, such regularization constrains the model's expressive capacity, potentially leading to a large $\hat{L}_\gamma(f)$ and increasing the population risk. This observation is empirically supported by experiments in Appendix D.7 (see Table 10).

**V. GRDL has a tighter bound than GIN with global pooling** In Appendix A, we provide the generalization error bound, i.e., Theorem A.1, for GIN with global pooling and compare it with Theorem 3.2. The result shows that our GRDL has a stronger generalization ability than GIN, which is further supported by the numerical results in Table 4.

Table 1: Classification accuracy (%). Bold text indicates the top 3 mean accuracy.

| METHOD | DATASET | | | | | | | | AVERAGE |
|---|---|---|---|---|---|---|---|---|---|
| | MUTAG | PROTEINS | NCI1 | IMDB-B | IMDB-M | PTC-MR | BZR | COLLAB | |
| PATCHY-SAN | **92.6±4.2** | 75.1±3.3 | 76.9±2.3 | 62.9±3.9 | 45.9±2.5 | 60.0±4.8 | 85.6±3.7 | 73.1±2.7 | 71.5 |
| GIN | 89.4±5.6 | 76.2±2.8 | 82.2±0.8 | 64.3±3.1 | 50.9±1.7 | 64.6±7.0 | 82.6±3.5 | 79.3±1.7 | 73.6 |
| DROPGIN | 90.4±7.0 | 76.9±4.3 | 81.9±2.5 | 66.3±4.5 | 51.6±3.2 | 66.3±8.6 | 77.8±2.6 | 80.1±2.8 | 73.9 |
| DIFFPOOL | 89.4±4.6 | 76.2±1.4 | 80.9±0.7 | 61.1±3.0 | 45.8±1.4 | 60.0±5.2 | 79.8±3.6 | 80.8±1.6 | 71.8 |
| SEP | 89.4±6.1 | 76.4±0.4 | 78.4±0.6 | **74.1±0.6** | 51.5±0.7 | 68.5±5.2 | 86.9±0.8 | **81.3±0.2** | 75.8 |
| GMT | 89.9±4.2 | 75.1±0.6 | 79.9±0.4 | **73.5±0.8** | 50.7±0.8 | **70.2±6.2** | 85.6±0.8 | 80.7±0.5 | 75.7 |
| MINCUTPOOL | 90.6±4.6 | 74.7±0.5 | 74.3±0.9 | 72.7±0.8 | 51.0±0.7 | 68.3±4.4 | 87.2±1.0 | **80.9±0.3** | 75.0 |
| ASAP | 87.4±5.7 | 73.9±0.6 | 71.5±0.4 | 72.8±0.5 | 50.8±0.8 | 64.6±6.8 | 85.3±1.3 | 78.6±0.5 | 73.1 |
| WITTOPOPOOL | 89.4±5.4 | 80.0±3.2 | 79.9±1.3 | 72.6±1.8 | **52.9±0.8** | 64.6±6.8 | 87.8±2.4 | 80.1±1.6 | 75.9 |
| OT-GNN | 91.6±4.6 | 76.6±4.0 | **82.9±2.1** | 67.5±3.5 | 52.1±3.0 | 68.0±7.5 | 85.9±3.3 | 80.7±2.9 | 75.7 |
| WEGL | 91.0±3.4 | 73.7±1.9 | 75.5±1.4 | 66.4±2.1 | 50.3±1.0 | 66.2±6.9 | 84.4±4.6 | 79.6±0.5 | 73.4 |
| FGW - ADJ | 82.6±7.2 | 72.4±4.7 | 74.4±2.1 | 70.8±3.6 | 48.9±3.9 | 55.3±8.0 | 86.9±1.0 | 80.6±1.5 | 71.5 |
| FGW - SP | 84.4±7.3 | 74.3±3.3 | 72.8±1.5 | 65.0±4.7 | 47.8±3.8 | 55.5±7.0 | 86.9±1.0 | 77.8±2.4 | 70.6 |
| WL | 87.4±5.4 | 74.4±2.6 | **85.6±1.2** | 67.5±4.0 | 48.4±4.2 | 56.0±3.9 | 81.3±0.6 | 78.5±1.7 | 72.4 |
| WWL | 86.3±7.9 | 73.1±1.4 | **85.7±0.8** | 71.6±3.8 | 52.6±3.0 | 52.6±6.8 | 87.6±0.6 | **81.4±2.1** | 73.9 |
| SAT | **92.6±4.3** | 77.7±3.2 | 82.5±0.8 | 70.0±1.3 | 47.3±3.2 | 68.3±4.9 | **91.7±2.1** | 80.6±0.6 | 76.1 |
| GRAPHORMER | 89.6±6.2 | 76.3±2.7 | 78.6±2.1 | 70.3±0.9 | 48.9±2.0 | **71.4±5.2** | 85.3±2.3 | 80.3±1.3 | 75.1 |
| GRDL | **92.1±5.9** | **82.6±1.2** | 80.4±0.8 | **74.8±2.0** | **52.9±1.8** | 68.3±5.4 | **92.0±1.1** | 79.8±0.9 | **77.9** |
| GRDL-W | 90.8±4.6 | **82.1±0.9** | 80.9±0.8 | 72.2±3.1 | **53.1±0.9** | 68.5±3.2 | 90.6±1.5 | 80.4±1.1 | **77.3** |
| GRDL-S | 90.6±5.7 | **81.1±1.4** | 81.2±1.5 | 72.4±3.3 | 52.5±1.1 | 64.2±3.2 | 91.6±1.3 | 78.6±1.3 | 76.5 |

***Remark*** 3.4. Currently, we use $K$ reference distributions for classification (one for each class). One natural approach to enhancing the model's expressive power is increasing the number of references for each class. However, counterintuitively, our empirical observations, supported by Theorem B.1, suggest that having only one reference per class is optimal. We discuss this further in Appendix B.

# 4   Related Work

Various sophisticated pooling operations have been designed to preserve the structural information of graphs [Bianchi et al., 2020, Ranjan et al., 2020, Baek et al., 2021, Chen and Gel, 2023, Yu et al., 2024]. For instance, DIFFPOOL, designed by Ying et al. [2018], learns a differentiable soft cluster assignment for nodes and maps nodes to a set of clusters to output a coarsened graph. Another method by Lee et al. [2019] utilizes a self-attention mechanism to distinguish nodes for retention or removal, and both node features and graph topology are considered with the self-attention mechanism.

A recent research direction focuses on preserving structural information by leveraging the optimal transport (OT) [Peyré and Cuturi, 2020]. OT-GNN, proposed by Chen et al. [2021], embeds a graph to a vector by computing Wasserstein distances between node embeddings and some "learned point clouds". TFGW, introduced by Vincent-Cuaz et al. [2022], embeds a graph to a vector of Fused Gromov-Wasserstein (FGW) distance [Vayer et al., 2018] to a set of "template graphs". OT distances have also been combined with dictionary learning to learn graph vector embedding in an unsupervised way (GDL) [Liu et al., 2022a, Vincent-Cuaz et al., 2021, Zeng et al., 2023].

Similar to the "learned point clouds" in OT-GNN, "template graphs" in TFGW, and dictionaries in GDL, our GRDL preserves information in node embeddings using reference distributions. To the best of the authors' knowledge, we are the first to model a graph's node embeddings as a discrete distribution and propose to classify it directly without aggregating it into a vector, marking our novel contribution. Additionally, our work stands out as the first to analyze the generalization bounds for this type of model, adding a theoretically grounded dimension to the research. By the way, our method is much more efficient than OT-GNN and TFGW. Please see Figure 2 and Table 8.

# 5   Numerical Experiments

## 5.1   Graph Classification Benchmark

**Datasets**   We leverage eight popular graph classification benchmarks [Morris et al., 2020], comprising five bioinformatics datasets (MUTAG, PROTEINS, NCI1, PTC-MR, BZR) and three social network datasets (IMDB-B, IMDB-M, COLLAB). We also use three large-scale imbalanced datasets (PC-3, MCF-7, and ogbg-molhiv [Hu et al., 2020]). A summary of data statistics is in Table 6.

**Baselines** Our approach is benchmarked against four groups of state-of-the-art baselines: 1) GNN models with global or sophisticated pooling operations, including PATCHY-SAN [Niepert et al., 2016], DIFFPOOL [Ying et al., 2018], GIN [Xu et al., 2018], DropGIN [Papp et al., 2021], SEP [Wu et al., 2022], GMT [Baek et al., 2021], MinCutPool [Bianchi et al., 2020], ASAP [Ranjan et al., 2020], and Wit-TopoPool

Table 2: AUC-ROC scores of large imbalanced data classification. Bold text indicates the best.

| METHOD | DATASET | | |
|---|---|---|---|
| | PC-3 | MCF-7 | OGBG-MOLHIV |
| GIN | 84.6±1.4 | 80.6±1.5 | 77.8±1.3 |
| DIFFPOOL | 83.2±1.9 | 77.2±1.3 | 73.7±1.8 |
| PATCHY-SAN | 80.7±2.1 | 78.9±3.1 | 70.2±2.1 |
| GRDL | **85.1±1.6** | **81.4±1.3** | **79.8±1.0** |

[Chen and Gel, 2023]; 2) Optimal transport based models such as WEGL [Kolouri et al., 2020] and OT-GNN [Chen et al., 2021]; 3) Kernel-based approaches including FGW [Titouan et al., 2019] operating on adjacency (ADJ) and shortest path (SP) matrices, the WL subtree kernel [Shervashidze et al., 2011], and the Wasserstein WL kernel [Togninalli et al., 2019]; 4) Graph transformers including Graphormer [Ying et al., 2021] and SAT [Chen et al., 2022a]. We also show the results of two variations of our GRDL: GRDL using Sinkhorn divergence (GRDL-S) and GRDL using Wasserstein distance (GRDL-W). For large imbalanced datasets, we only benchmark our GRDL against PATCHY-SAN, GIN, and DIFFPOOL because other methods are too costly. Details about the initialization and hyper-parameters setting can be found in Appendix D.3.

**Experiment Settings** Due to the page limitation, please refer to Appendix D.2.

**Classification Results** Table 1 shows the classification results. The AUC-ROC scores of experiments results on the three large imbalanced datasets are reported in Table 2. Our method has top 3 classification performance over baselines in almost all datasets. Our GRDL, GRDL-W and GRDL-S have close performance. However, as shown later in Figure 2, our original GRDL has significantly lower time costs and thus is preferable for practical use. Graph transformers also have competitive performance, but they have significantly larger amount parameters and much higher time costs than our model, as shown by Table 13 in Appendix D.8 .

## 5.2 Time Cost Comparison

We compare the time cost of our GRDL with two models that leverage optimal transport distances discussed in Section 4: OT-GNN [Chen et al., 2021] and TFGW [Vincent-Cuaz et al., 2022]. Compared with them, our model has significantly lower time costs. We present empirical average training time per epoch in Figure 2 and average prediction time per graph in Table 9 in Appendix D.4. Experiments were conducted on CPUs (Apple M1) using identical batch sizes, ensuring a fair comparison. It's noteworthy that the OT solver employed in TFGW and OT-GNN is currently confined to CPU, influencing the choice of hardware for this evaluation. We analyzed the theoretical time complexity in Appendix D.4 (see Table 8).

Table 3: Comparison of time cost (second) per epoch with Wit-TopoPool and MSGNN.

| | MUTAG | PROTEINS | NCI1 | IMDB-B | IMDB-M | PTC-MR | BZR | COLLAB | SYN-100 | SYN-300 | SYN-500 |
|---|---|---|---|---|---|---|---|---|---|---|---|
| GRDL (OURS) | 0.4 | 3.4 | 12.6 | 2.4 | 3.5 | 0.8 | 1.2 | 16.3 | 26.6 | 45.8 | 88.7 |
| WITTOPOPOOL | 0.4 | 2.6 | 21.4 | 2.4 | 2.6 | 1.0 | 1.3 | 39.1 | 32.9 | 50.8 | 97.5 |
| MSGNN | 45.2 | - | - | - | - | 75.5 | 135.3 | - | - | - | - |

We also compare training time with two latest pooling methods including Wit-TopoPool [Chen and Gel, 2023] and MSGNN [Lv et al., 2023] on eight real datasets and three synthetic datasets. The three synthetic datasets have 2000 graphs with 100(SYN-100), 300(SYN-300), and 500(SYN-500) nodes per graph, respectively. The edge number is $0.1n^2$ where $n$ is the number of nodes. The empirical training time per epoch is shown in Table 3, where empty of MSGNN means it takes more than 200 seconds to train a single epoch, which is too costly. As can be seen, our method is the most efficient among these three methods.

## 5.3 Graph Visualization

We use t-SNE [Van der Maaten and Hinton, 2008] to visualize the distributions of graphs' node embeddings given by our GRDL model, which is equivalent to visualizing each graph in a 3-D coordinate system. Firstly we use MMD to calculate a distance matrix $\mathbf{C} \in \mathbb{R}^{(N+K)\times(N+K)}$ between

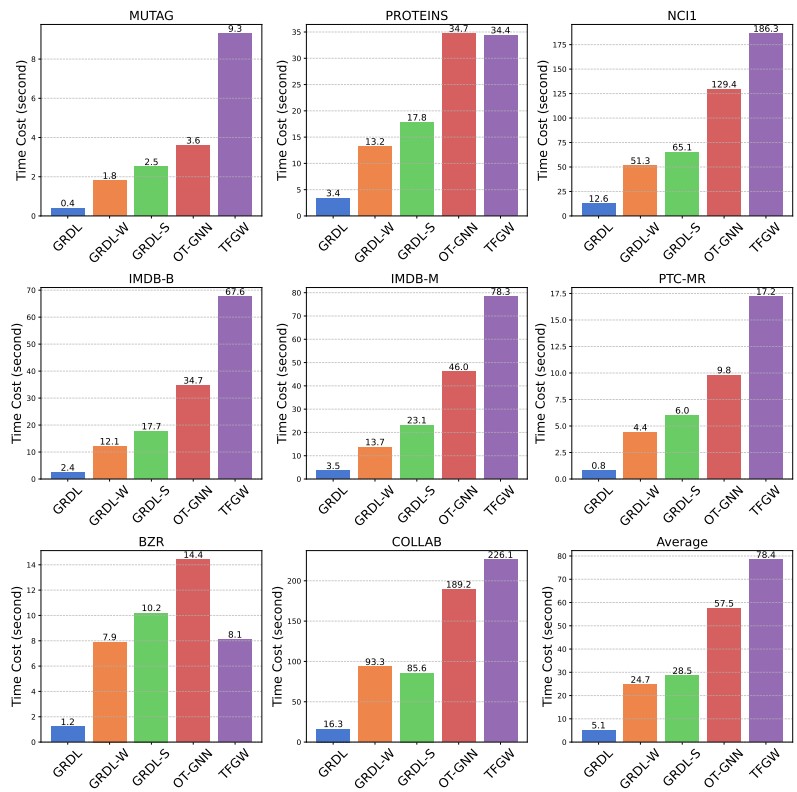

Figure 2: Average training time per epoch. GRDL is 10 times faster than OT-GNN and TFGW.

the node embeddings $\{\mathbf{H}_i\}_{i=1}^N$ and the reference distributions $\{\mathbf{D}_k\}_{k=1}^K$. The 3-D visualization given by t-SNE using $\mathbf{C}$ is presented in Figure 3. The graphs are located around the references. It means that the learned references can represent realistic graphs' latent node embeddings from the data.

## 5.4 More Numerical Results

The ablation study, influence of $\theta$, generalization comparison with GIN are in Appendices D.5, D.6, and A, respectively.

## 6 Conclusions

We proposed GRDL, a novel framework for graph classification without global pooling operations and hence effectively preserve the information of node embeddings. What's more, we theoretically analyzed the generalization ability of GRDL, which provided valuable insights into how the generalization ability scales with the properties of the graph data and network structure. Extensive experiments on moderate-scale and large-scale benchmark datasets verify the effectiveness and efficiency of GRDL in comparison to baselines. However, on some benchmark datasets (e.g. NCI1), our model does not outperform the baseline, which may be a limitation of our work and requires further investigation in the future.

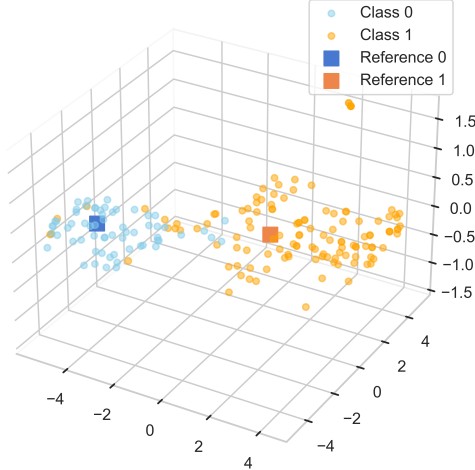

Figure 3: T-SNE visualization of MUTAG embeddings and reference distributions given by GRDL. Each dot denotes a graph and each square denotes a reference distribution.

## Acknowledgments

This work was supported by the National Natural Science Foundation of China under Grant No.62376236, the Guangdong Provincial Key Laboratory of Mathematical Foundations for Artificial Intelligence (2023B1212010001), Shenzhen Science and Technology Program ZDSYS20230626091302006, Shenzhen Stability Science Program 2023, and Hetao Shenzhen-Hong Kong Science and Technology Innovation Cooperation Zone Project (No.HZQSWS-KCCYB-2024016). The authors declare that they have no known competing financial interests or personal relationships that could have appeared to influence the work reported in this paper.

The authors appreciate the reviewers and area chair of the paper.

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

# Appendix

## Table of Contents

# A  Generalization Comparison to GIN with Global Pooling

To see the advantage of our GRDL, we compare it with GIN. The only difference between GRDL and GIN is that GRDL uses a reference layer while GIN uses readout. We add an $r'$-layer MLP as the classifier after message-passing modules in the GIN. The following theorem gives an upper bound of GIN's generalization error:

**Theorem A.1** (Generalization bound of GIN). *Let $n = \min_i n_i$, $c = \|\tilde{\mathbf{A}}\|_\sigma$, and $\bar{d} = \max_{i,l} d_i^{(l)}$. Denote $R_G := c^{2L}\|\mathbf{X}\|_2^2 \ln(2\bar{d}^2)\big(\prod_{l=1}^L(\prod_{i=1}^r \kappa_i^{(l)})^2\big)\big(\sum_{l=1}^L \sum_{i=1}^r \big(\frac{b_i^{(l)}}{\kappa_i^{(l)}}\big)^{2/3}\big)^3$. For graphs $\mathcal{G} = \{(G_i, y_i)\}_{i=1}^N$ drawn i.i.d from any probability distribution over $\mathcal{X} \times \{1, \ldots, K\}$, with probability at least $1 - \delta$, GIN network with mean readout satisfies*

$$L_\gamma(f) \le \hat{L}_\gamma(f) + 3\gamma\sqrt{\frac{\ln(2/\delta)}{2N}} + \frac{8\gamma + 24\mu(\prod_{i=1}^{r'} \kappa_i^{(L+1)})\sqrt{R_G + R_G'}\ln N}{N}$$

*where $R_G' = c^{2L}\|\mathbf{X}\|_2^2 \ln(2\bar{d}^2)\big(\prod_{l=1}^L(\prod_{i=1}^r \kappa_i^{(l)})^2\big)\big(3C_2^2 C_1 + 3C_2 C_1^2 + C_1^3\big)$, $C_1 = \sum_{i=1}^{r'} \frac{b_i^{(L+1)}}{\kappa_i^{(L+1)}}$, and $C_2 = \sum_{l=1}^L \sum_{i=1}^r \big(\frac{b_i^{(l)}}{\kappa_i^{(l)}}\big)^{2/3}$.*

This bound is derived using the same techniques as our GRDL bound in Theorem 3.2. To compare these two bounds, we only need to compare the following two terms:

$$Q_{\text{GRDL}} := 24\sqrt{v_1 + v_2}\ln N + 24\gamma\sqrt{N v_2 \ln v_3} \qquad \text{(Theorem 3.2)}$$

$$Q_{\text{GIN}} := 24\mu\Big(\prod_{i=1}^{r'} \kappa_i^{(L+1)}\Big)\sqrt{R_G + R_G'}\ln N \qquad \text{(Theorem A.1)}$$

where $v_1 = \frac{64\theta K\mu^2}{n}R_G$, $v_2 = Km\bar{d}$, and $v_3 = \frac{24\sqrt{\theta N}b_D\mu}{\sqrt{m}}$. Our observations are as follows.

- The $\theta$ in $v_1$ can be absorbed into $\mathbf{W}_r^{(L)}$ and $\{\mathbf{D}_k\}_{k=1}^K$. Since $n \gg K$, we conclude that $\frac{64\theta K}{n}$ is smaller than 1 in practice. Therefore, $v_1 \le \mu^2 R_G$.
- $v_2 = Km\bar{d}$ and $v_3 = \frac{24\sqrt{\theta N}b_D\mu}{\sqrt{m}}$ are much smaller than $R_G'$ as well as $R_G$, i.e., $v_2 \ll R_G'$ and $v_3 \ll R_G'$. The reason is that $R_G'$ and $R_G$ involve the multiplication of terms related to $c$, $\|\mathbf{X}\|_2^2$, and $\kappa_i^{(l)}$.
- In Theorem A.1, $\prod_{i=1}^{r'} \kappa_i^{(L+1)}$ is typically larger than 3 for $r' > 1$ based on empirical observations. We also observe that $\prod_{i=1}^{r'} \kappa_i^{(L+1)}$ may be smaller than 1 for $r' = 1$, but the linear classifier's expressive capacity is very limited and thus has large training error. Therefore, we focus on the case where $r' > 1$.

Now we can conclude that $Q_{\text{GRDL}} < Q_{\text{GIN}}$. Therefore, the generalization error upper bound of GIN is larger than of GRDL, meaning our GRDL generalizes better on unseen data than GIN in the worst case. It is worth noting that these results apply to other GNNs such as GCN.

We now use numerical experiments on real datasets to support our claim. The training and testing accuracy of GRDL and GIN are shown in Table 4. We see that the training accuracy of our GRDL is close to that of GIN, but the testing accuracy of our GRDL is much higher than that of GIN. This means that GRDL and GIN have similar training errors but the former has a stronger generalization ability.

# B  Theory and Experiments of GRDL with Multiple Reference Distributions

Currently, we use $K$ reference distributions for classification (one for each class). One natural approach to enhance the model's expressive power is to increase the number of reference distributions for each class. However, counterintuitively, our empirical observations suggest that having only one reference per class is optimal. In this section, we will explore and provide insights into this phenomenon.

Table 4: Comparison of the trainining and testing accuracy between GRDL and GIN.

| Dataset | GRDL | | GIN | |
| | Training Accuracy | Testing Accuracy | Training Accuracy | Testing Accuracy |
|---|---|---|---|---|
| MUTAG | 93.3 | 92.1 | 92.7 | 89.4 |
| PROTEINS | 83.1 | 82.6 | 80.5 | 76.2 |
| NCI1 | 82.8 | 80.4 | 83.3 | 82.2 |
| IMDB-B | 76.3 | 74.8 | 75.9 | 64.3 |
| IMDB-M | 53.1 | 52.9 | 51.7 | 50.9 |
| PTC-MR | 71.3 | 68.3 | 66.1 | 64.6 |
| BZR | 93.1 | 92.0 | 93.9 | 82.6 |
| COLLAB | 82.1 | 79.9 | 82.3 | 79.3 |
| Average | 79.4 | 77.9 | 78.3 | 73.6 |

Suppose we have $P$ reference distributions for each class, i.e. $\mathcal{D} \triangleq \left\{ \mathbf{D}_k^{(p)} \right\}_{k \in [K]}^{p \in [P]}$, where $\mathbf{D}_k^{(p)}$ is the $p$-th reference in the $k$-th class. The prediction in Equation (5) is changed to

$$y_{\text{pred},i} = \arg \max_{k \in [K]} s_{ik}, \quad s_{ik} = \sum_{p=1}^{P} \xi(\mathbf{H}_i, \mathbf{D}_k^{(p)}). \tag{24}$$

The training algorithm is nearly the same as our GRDL with one reference per class (Algorithm 1) except for the mini-batch training loss because of the multiple references, i.e.,

$$\mathcal{L} = -\frac{1}{B} \sum_{i \in \mathcal{B}} \sum_{k=1}^{K} y_{ik} \log \hat{y}_{ik} + \lambda \sum_{k' \neq k} \sum_{p,p'=1}^{P} \xi_\theta(\mathbf{D}_k^{(p)}, \mathbf{D}_{k'}^{(p')}) \tag{25}$$

We compare the model with $P = 2$ (GRDL-2) and $P = 3$ (GRDL-3) with our GRDL ($P = 1$). Table 5 shows the classification accuracy of the models on the benchmark datasets.

Table 5: Classification accuracy of models with multiple reference distributions. **Bold** text indicates the best mean accuracy.

| DATASET | METHOD | | |
| | GRDL | GRDL-2 | GRDL-3 |
|---|---|---|---|
| MUTAG | **92.1±5.9** | 91.5±4.8 | 90.4±3.1 |
| PROTEINS | **82.6±1.2** | 81.4±2.1 | 81.3±2.9 |
| NCI1 | **80.4±0.8** | 79.3±1.0 | 80.0±1.6 |
| IMDB-B | **74.8±2.0** | 73.6±2.2 | 74.0±1.4 |
| IMDB-M | **52.9±1.8** | 51.1±1.2 | 50.3±2.1 |
| PTC-MR | **68.3±5.4** | 66.3±6.4 | 65.4±5.5 |
| BZR | **92.0±1.1** | 87.1±2.7 | 88.2±3.1 |
| COLLAB | **79.8±0.9** | 77.9±1.2 | 77.5±0.7 |

To explain why GRDL performs better than the models with more references, we first introduce the following theorem

**Theorem B.1.** *Let $n$ be the minimum number of nodes for graphs $\{G_i\}_{i=1}^{N}$, $\theta$ be the hyper-parameter in the Gaussian kernel (Equation (15)), $c = \|\tilde{\mathbf{A}}\|_\sigma$. For graphs $\mathcal{G} = \{(G_i, y_i)\}_{i=1}^{N}$ drawn i.i.d from any probability distribution over $\mathcal{X} \times \{1, \ldots, K\}$ and references $\left\{ \mathbf{D}_k^{(p)} \right\}_{k \in [K]}^{p \in [P]}, \mathbf{D}_k^{(p)} \in \mathbb{R}^{m \times d}$, with probability at least $1 - \delta$, every loss function $l_\gamma$ and network $f \in \mathcal{F}$ under Assumption 3.1 satisfy*

$$L_\gamma(f) \leq \hat{L}_\gamma(f) + 3\gamma \sqrt{\frac{\ln(2/\delta)}{2N}} + \frac{8\gamma + 24\sqrt{v_1 + v_2} \ln N + 24\gamma\sqrt{N}v_2 \ln v_3}{N}$$

*where*

$$v_1 = \frac{64\theta P^2 K R_G \mu^2}{n}, v_2 = Km\bar{d}, v_3 = \frac{24P\sqrt{\theta N}b_D\mu}{\sqrt{m}}, R_G = c^{2L}\|\mathbf{X}\|_2^2 \ln(2\bar{d}^2)\left(\prod_{l=1}^{L}\left(\prod_{i=1}^{r}\kappa_i^{(l)}\right)^2\right)\left(\sum_{l=1}^{L}\sum_{i=1}^{r}\left(\frac{b_i^{(l)}}{\kappa_i^{(l)}}\right)^{2/3}\right)^3.$$

This is essentially a more general version of Theorem 3.2. The following is a brief proof of this theorem.

*Proof.* The only difference between multiple reference distributions and a single reference distribution comes from the calculation of $s_{ij}$.

$$|s_{ij} - s'_{ij}| = \left|\sum_{p=1}^{P}\left(\text{MMD}^2\left(\mathbf{H}_i, \mathbf{D}_j^{(p)}\right) - \text{MMD}^2\left(\mathbf{H}'_i, \mathbf{D}_j^{(p)'}\right)\right)\right|$$

$$\leq \sum_{p=1}^{P}\left|\text{MMD}^2\left(\mathbf{H}_i, \mathbf{D}_j^{(p)}\right) - \text{MMD}^2\left(\mathbf{H}'_i, \mathbf{D}_j^{(p)'}\right)\right|$$

$$\leq 4\sqrt{\theta}P\left(n^{-1/2}\|\mathbf{H}_i - \mathbf{H}'_i\|_2 + m^{-1/2}\|\mathbf{D}_j - \mathbf{D}'_j\|_2\right)$$

Then with minor modifications of proof of Lemma F.14, the covering number of $\mathcal{F}$ is given by

$$\ln \mathcal{N}\left(\epsilon, \mathcal{F}, \rho\right) \leq \frac{64\theta P^2 K R_G}{n\epsilon^2} + Kmd\ln\left(\frac{24b_D P\sqrt{\theta N}}{\sqrt{m}\epsilon}\right).$$

Then the theorem can be proved using the same process as the proof of Theorem 3.2. $\square$

Let $\bar{\kappa} = \max_{i,l}\kappa_i^{(l)}$, $\bar{b} = \max_{i,l}\frac{b_i^{(l)}}{\kappa_i^{(l)}}$ and suppose $\delta$ is sufficiently large. The bound in Theorem B.1 can be simplified to

$$L_\gamma(f) \leq \hat{L}_\gamma(f) + \tilde{\mathcal{O}}\left(\frac{\sqrt{v_1} + \gamma\sqrt{N v_2}}{N}\right) \leq \hat{L}_\gamma(f) + \tilde{\mathcal{O}}\left(\frac{\mu\bar{b}\|\mathbf{X}\|_2 c^L(Lr)^{\frac{3}{2}}\bar{\kappa}^{Lr}P\sqrt{\theta K/n}}{N} + \gamma\sqrt{\frac{Kmd}{N}}\right)$$
$$(26)$$

Empirically, we observe that the training loss $\hat{L}_\gamma$ and the misclassification rate are nearly the same for small $P$ and large $P$ as shown in Figure 4. Therefore, smaller $P$ implies tighter generalization bound in (26). This means that one reference distribution for each class ($P = 1$) may be the optimal choice.

Another explanation is that, the nodes embeddings of graphs in the same class can be regarded as samples drawn from a single discrete distribution, thus learning a single reference distribution is sufficient to provide high classification accuracy. On the other hand, the union of multiple distributions can be regarded as a single distribution.

## C Detailed Training Algorithm of GRDL

In practice, since the scale of $\theta$ is different from the scale of other parameters in the model, a different learning rate is used to update it. Suppose Adam [Kingma and Ba, 2014] is used to optimize the parameters, then the training of GRDL model is presented in Algorithm 1.

## D More Experiments

### D.1 Dataset Statistics

The statistics of the datasets are reported in Table 6. PC-3, MCF-7, and ogbg-molhiv are three large graph datasets.

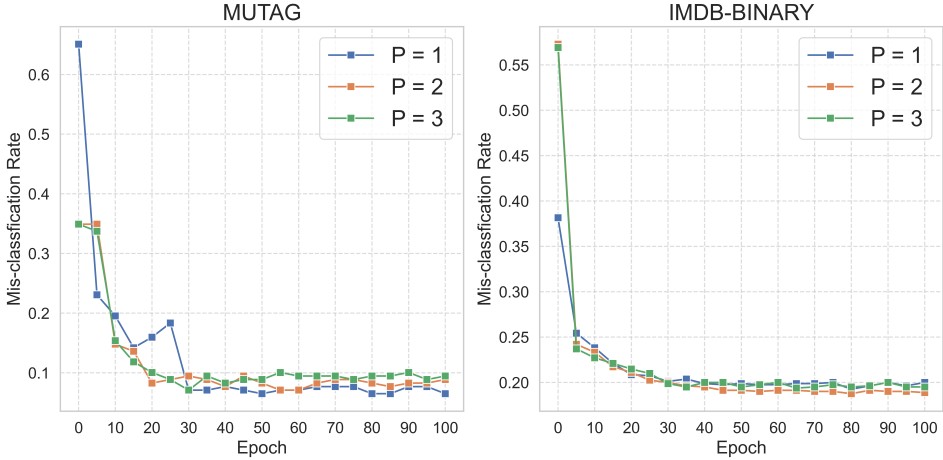

Figure 4: Training data misclassification rate on MUTAG (left) and IMDB-BINARY (right) with different numbers of references for each class ($P$). The effect of $P$ on the training misclassification rates is not obvious.

---

**Algorithm 1** GRDL Training

---

1: **Input:** Graphs $\{G_i\}_{i=1}^N$, $\alpha_1$ the learning rate of $\mathbf{w}$ and $\mathcal{D}$, $\alpha_2$ the learning rate of $\theta$, batch size $B$.
2: Initialize $\mathbf{w}$, $\mathcal{D}$, and $\theta$.
3: **repeat**
4:     Sample a minibatch $\{G_i : i \in \mathcal{B}, |\mathcal{B}| = B\}$
5:     $\mathbf{s}_i \leftarrow f_{\mathbf{w}, \mathcal{D}, \theta}(G_i)$, $i \in \mathcal{B}$
6:     $\hat{\mathbf{y}}_i \leftarrow \mathrm{softmax}(\mathbf{s}_i)$, $i \in \mathcal{B}$
7:     $\mathcal{L} = -\dfrac{1}{B} \sum\limits_{i \in \mathcal{B}} \sum\limits_{k=1}^{K} y_{ik} \log \hat{y}_{ik} + \lambda \sum\limits_{k' \neq k} \xi_\theta(\mathbf{D}_k, \mathbf{D}_{k'})$
8:     $(g_{\mathbf{w}}, g_{\mathcal{D}}, g_\theta) \leftarrow \nabla_{\mathbf{w}, \mathcal{D}, \theta} \mathcal{L}$
9:     $(\mathbf{w}, \mathcal{D}) \leftarrow (\mathbf{w}, \mathcal{D}) + \alpha_1 \cdot \mathrm{Adam}\left((\mathbf{w}, \mathcal{D}), (g_{\mathbf{w}}, g_{\mathcal{D}})\right)$
10:    $\theta \leftarrow \theta + \alpha_2 \cdot \mathrm{Adam}(\theta, g_\theta)$
11: **until** convergence conditions are met
12: **Output:** $f_{\mathbf{w}, \mathcal{D}, \theta}$

---

## D.2 Experiment Settings

In the GNN literature, researchers typically perform 10-fold cross-validation and report the best average accuracy along with standard deviation [Xu et al., 2018, Papp et al., 2021, Maron et al., 2019]. But here, we adopt a different strategy used in [Vincent-Cuaz et al., 2022]. We quantify the generalization capacities of models by performing a 10-fold cross-validation with a holdout test set which is never seen during training. The validation accuracy is tracked every 5 epochs, and the model that maximizes the validation accuracy is retained for testing. This setting is more realistic than a simple 10-fold CV and can better quantify models' generalization performances [Bengio and Grandvalet, 2003]. However, the test sets for MUTAG and PTC-MR contain only 18 and 34 graphs respectively, making them too small for assessing generalization ability. Therefore, for MUTAG and PTC-MR, we use 10-fold cross-validation following [Xu et al., 2018]. Notice that kernel-based methods do not require a stopping criterion dependent on a validation set, so we report results of 10-fold nested cross-validation repeated 10 times following [Titouan et al., 2019].

## D.3 Hyper-parameter Settings and Parameter Initializations

**Model** For all the baselines, we adopt the hyper-parameters suggested in the original papers. For our methods, we use GIN [Xu et al., 2018] layers as the embedding network. Every GIN layer is an

Table 6: Dataset Statistics

| datasets | features | #graphs | #classes | min #nodes | median #nodes | max #nodes |
|---|---|---|---|---|---|---|
| MUTAG | $\{0,\ldots,6\}$ | 188 | 2 | 10 | 17 | 28 |
| PROTEINS | $\mathbb{R}^{29}$ | 1113 | 2 | 2 | 13 | 63 |
| NCI1 | $\{0,\ldots,36\}$ | 4110 | 2 | 3 | 27 | 111 |
| PTC-MR | $\{0,\ldots,17\}$ | 344 | 2 | 2 | 13 | 64 |
| BZR | $\mathbb{R}^3$ | 405 | 2 | 13 | 35 | 57 |
| IMDB-B | None | 1000 | 2 | 12 | 17 | 136 |
| IMDB-M | None | 1500 | 3 | 7 | 10 | 89 |
| COLLAB | None | 5000 | 3 | 32 | 52 | 492 |
| PC-3 | $\{0,\ldots,28\}$ | 27509 | 2 | 3 | 24 | 113 |
| MCF-7 | $\{0,\ldots,28\}$ | 27770 | 2 | 3 | 24 | 113 |
| ogbg-molhiv | $\mathbb{R}^{29}$ | 41127 | 2 | 2 | 23 | 222 |

MLP of 2 layers ($r = 2$) with batch normalization, whose number of units is validated in $\{32, 64\}$ for all datasets. The parameter $\lambda$ is validated in $\{0.1, 1\}$. We validate the number of GIN layers ($L$) in $\{3, 4, 5, 6, 7, 8, 9\}$. For each dataset, the references' dimension ($m$) is validated in the minimum number of nodes (G1), average of the minimum and median number of nodes (G2), median number of nodes (G3), average of the median and maximum number of nodes (G4), and the maximum number of nodes (G5) of graphs in the dataset. The reference for each class is initialized as follows: 1) If there is a $m$-node graph in the dataset that belongs to the corresponding class, then the reference is initialized as node embeddings of the graph. 2) If no graph in the class has $m$ nodes, then we perform K-Means clustering on the graphs of the class, and the $m$ clustering center is chosen to be the initial reference.

**Optimization** The models are trained with Adam optimizer with an initial learning rate $\alpha_1 = 10^{-3}$ for network weights and references. The learning rate $\alpha_1$ decays exponentially with a factor 0.95. Since the Gaussian kernel parameter $\theta$ is small in practice (around $10^{-3}$ in our model), it is hard to choose a learning rate for it. Therefore, we consider $\pi = \frac{1}{\theta}$ instead. $\pi$ is initialized to $500$ and its initial learning rate $\alpha_2$ is validated in $\{0.1, 1\}$. The batch size for all datasets is fixed to 32.

Detailed hyper-parameters setting can be found in Table 7.

.

Table 7: Hyper-parameter settings for experiment results in Table 1

| DATASETS | #LAYERS | REFERENCE DIM | HIDDEN CHANNELS | $\lambda$ | $\alpha_2$ |
|---|---|---|---|---|---|
| MUTAG | 5 | G2 | 32 | 1.0 | 1.0 |
| PROTEINS | 5 | G3 | 32 | 1.0 | 0.1 |
| NCI1 | 5 | G3 | 32 | 1.0 | 0.1 |
| IMDB-B | 5 | G3 | 32 | 1.0 | 1.0 |
| IMDB-M | 5 | G3 | 32 | 0.1 | 0.1 |
| PTC-MR | 5 | G3 | 32 | 1.0 | 0.1 |
| BZR | 5 | G3 | 32 | 1.0 | 0.1 |
| COLLAB | 6 | G3 | 32 | 1.0 | 0.1 |
| PC-3 | 6 | G4 | 64 | 1.0 | 0.1 |
| MCF-7 | 6 | G4 | 64 | 1.0 | 0.1 |
| OGBG-MOLHIV | 6 | G4 | 64 | 1.0 | 0.1 |

## D.4 Time Complexity

To provide a comprehensive understanding, we first show the forward propagation time complexity of the three models for a single graph $G = (\mathbf{A}, \mathbf{X}), \mathbf{A} \in \mathbb{R}^{n \times n}, \mathbf{X} \in \mathbb{R}^{n \times d}$. Given that all three models employ a Graph Neural Network (GNN) for obtaining node embeddings, we denote the complexity and the number of parameters of the GNN embedding part as $C_1$ and $N_1$, respectively. Additionally, both OT-GNN and TFGW are augmented with an MLP for classification, introducing

an extra complexity $C_2$ and additional parameters $N_2$. In the case of our GRDL, the number of references aligns with the number of classes $K$.

Let's assume the number of references for OT-GNN is $K_1$, and for TFGW, it is $K_2$. Notably, TFGW and OT-GNN usually choose $K_1 = 2K$ and $K_2 = 2K$. Additionally, assuming that all references $\mathbf{D}_i \in \mathbb{R}^{m \times d}$. Since the GW distance is iteratively computed in Vincent-Cuaz et al. [2022], we denote the number of iterations for convergence as $T$. The time complexity and the number of parameters are outlined in Table 8.

Table 8: Time complexity and number of parameters for GRDL, OT-GNN and TFGW.

| MODEL | COMPLEXITY | PARAMETERS |
|---|---|---|
| GRDL | $C_1 + \mathcal{O}\left(K(n^2 + mn + m^2)\right)$ | $N_1 + Kmd$ |
| OT-GNN | $C_1 + \mathcal{O}\left(K_1(m+n)^3 \log(m+n)\right) + C_2$ | $N_1 + K_1md + N_2$ |
| TFGW | $C_1 + \mathcal{O}\left(TK_2(m^2n + n^2m)\right) + C_2$ | $N_1 + K_2md + N_2$ |

Notice that Wasserstein distance can be approximated by sinkhorn iterations with complexity $O((m+n)^2)$ per iteration [Cuturi, 2013]. But in practice, the exact calculation with $O((m+n)^3 \log(m+n))$ complexity empirically gives better performance in terms of both precision and speed, so it is implemented in the original paper of OT-GNN [Chen et al., 2021]. Theoretically, our GRDL has lower prediction time complexity and a reduced parameter count. Table 9 shows the empirical prediction time per graph

Table 9: Average prediction time per graph ($10^{-3}$ seconds).

| DATASET | METHOD | | |
|---|---|---|---|
| | GRDL | OT-GNN | TFGW |
| MUTAG | **1.2** | 25.6 | 53.5 |
| PROTEINS | **1.3** | 29.4 | 48.3 |
| NCI1 | **1.2** | 25.1 | 83.5 |
| IMDB-B | **1.2** | 16.3 | 66.0 |
| IMDB-M | **1.1** | 16.3 | 66.5 |
| PTC-MR | **1.8** | 15.0 | 58.4 |
| BZR | **2.1** | 23.8 | 76.2 |
| AVERAGE | **1.4** | 21.6 | 64.6 |

Table 10: Comparison of GRDL with/without regularization on references norm.

| DATASET | METHOD | |
|---|---|---|
| | GRDL | GRDL-R |
| MUTAG | **92.1±5.9** | 91.6±5.5 |
| PROTEINS | **82.6±1.2** | 80.3±1.2 |
| NCI1 | **80.4±0.8** | 78.6±0.9 |
| IMDB-B | **74.8±2.0** | 73.6±1.6 |
| IMDB-M | **52.9±1.8** | 48.3±1.6 |
| PTC-MR | **68.3±5.4** | 68.1±6.4 |
| BZR | **92.0±1.1** | 90.7±2.4 |
| COLLAB | **79.8±0.9** | 77.9±0.7 |

## D.5 Ablation Study

We consider two variants of GRDL. The first one is GRDL with a fixed $\theta = 10^{-2}$ in the Gaussian kernel. The other is GRDL with $\lambda = 0$ in (9), which does not have the discriminativeness constraint on the references. We also include another baseline where we first use sum pooling over node embeddings and get the graph embedding vectors. The graph vectors are then used to compare with references (vectors in this case) with discrimination loss. The classification results of benchmark datasets are given in Table 11. The original model with learnable $\theta$ and discriminativeness constraints consistently outperforms the other two.

## D.6 Influence of $\theta$

In our model, we initialized the Gaussian kernel hyper-parameter $\theta$ to $2 \times 10^{-3}$ and it was adaptively learned during training. Actually, all values between $1 \times 10^{-4}$ and $1 \times 10^{-1}$ give similar performance, as it is adaptively adjusted in the training. The initialization of the neural network parameters and the reference distributions cannot guarantee that $\mathbf{x}$ is close to $\mathbf{x}'$. If $\theta$ is too large, the Gaussian kernel $k(\mathbf{x}, \mathbf{x}') = \exp\left(-\theta\|\mathbf{x} - \mathbf{x}'\|_2^2\right)$ will be too sharp, which will lead to almost zero values. Hence, MMD will fail to effectively quantify the distance between the embeddings and reference distributions, as shown in Table 12.

Table 11: Classification accuracy of ablation methods. Bold text indicates the best mean accuracy.

| DATASET | METHOD | | | |
|---|---|---|---|---|
| | GRDL | GRDL FIXED $\theta$ | GRDL $\lambda = 0$ | GRDL+SUM $\lambda = 0$ |
| MUTAG | **92.1±5.9** | 90.4±6.4 | 89.9±4.9 | 89.9±6.0 |
| PROTEINS | **82.6±1.2** | 81.8±0.9 | 81.8±1.3 | 78.4±0.6 |
| NCI1 | **80.4±0.8** | 80.2±2.2 | 80.0±1.6 | 77.2±1.7 |
| IMDB-B | **74.8±2.0** | 72.8±1.8 | 73.1±1.5 | 71.6±5.2 |
| IMDB-M | **52.9±1.8** | 52.1±1.2 | 51.3±1.4 | 49.8±5.4 |
| PTC-MR | **68.3±5.4** | 66.6±5.7 | 66.6±5.9 | 62.5±6.3 |
| BZR | **92.0±1.1** | 90.1±1.5 | 89.5±2.3 | 85.3±1.5 |
| COLLAB | **79.8±0.9** | 79.5±0.7 | 79.0±1.0 | 77.1±0.9 |

Table 12: Classification accuracy of MUTAG dataset with different $\theta$.

| $\theta$ | $1 \times 10^{-4}$ | $1 \times 10^{-3}$ | $1 \times 10^{-2}$ | $1 \times 10^{-1}$ | 1 | $1 \times 10^{1}$ | $1 \times 10^{2}$ | $1 \times 10^{3}$ |
|---|---|---|---|---|---|---|---|---|
| Accuracy | 0.9096 | 0.9149 | 0.9113 | 0.9113 | 0.8254 | 0.6822 | 0.5737 | 0.3345 |

## D.7 Experiments on The Generalization Error Bound

**Use moderate-size message passing network**   We choose the training loss to be the cross-entropy loss ($\lambda = 0$) and $l_\gamma$ is chosen the same as the training loss. We validate the number of GIN layers $L \in \{3, 4, 5, 6\}$ and the number of MLP layers for each GIN layer $r \in \{2, 3, 4\}$. $L_\gamma$ is set to be the validation loss and $\hat{L}_\gamma$ is set to be the training loss. From Figure 5, we have the following observations

- For any fixed $r$, if $L$ is increasing, the empirical risk $\hat{L}_\gamma$ increases, and the population risk $L_\gamma$ either increases ($r = 3$ and $r = 4$) or first decreases then increases ($r = 2$).
- For any fixed $L$, if $r$ is increasing, the population risk $L_\gamma$ first decreases then increases in most of the cases, and the empirical risk $\hat{L}_\gamma$ decreases in most of the cases.

These observations align with our bound and support our claim that moderate-size GNN should be used.

**Use moderate-size references**   We validate the reference size ($m$) in our experiments on real datasets, as detailed in Appendix D.3. The results in Appendix D.3 show that moderate-size references (G2, G3, G4) provide better generalization results. Here, we present the classification results for MUTAG and PROTEINS by choosing $m$ from more fine-grained sets. For MUTAG, $m$ is chosen from $1, 2, \ldots, 30$. For PROTEINS, $m$ is chosen from $1, 3, 5, \ldots, 61$. The figures in Figure 6 show that our model performs the best when a moderate $m$ is used.

**Regularization on references norm barely helps**   Consider a model regularizing the norm of references $\|\mathbf{D}\|_2$ (GRDL-R)

$$\min \mathcal{L}_{\text{CE}} + \lambda_1 \mathcal{L}_{\text{Dis}} + \lambda_2 \|\mathbf{D}\|_2 \tag{27}$$

The hyper-parameter $\lambda_2$ is set as $0.01$. Table 10 compares the empirical results of GRDL model and the results of GRDL-R model. From the table, we can see that GRDL performs better than GRDL-R on all datasets, which verifies our discussion in Section 3.3. Therefore, the regularization of references barely helps in our model.

## D.8 Parameter Number and Time Cost Comparison with Graph Transformers

Table 13 compares the number of parameters and training time per epoch of our GRDL with two graph transformers method. We can see that our GRDL has significantly fewer parameters and training time, making it preferable. All experiments are conducted on one RTX3080.

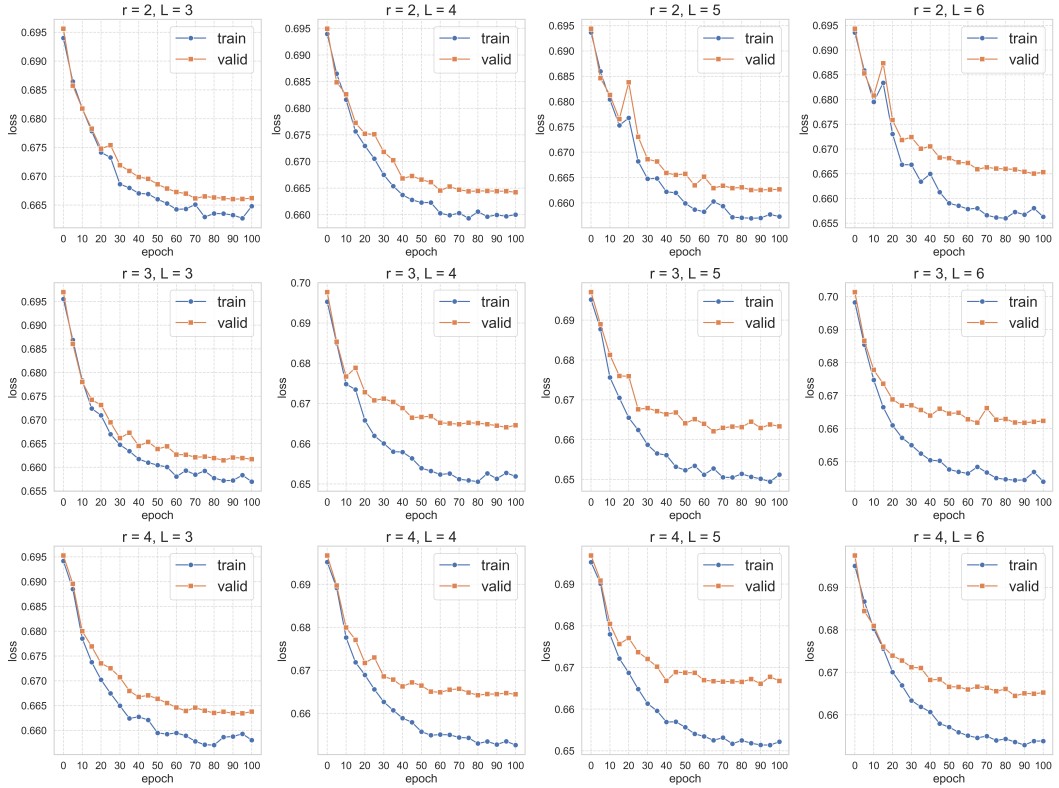

Figure 5: The blue and orange lines denote the training error $\hat{L}_\gamma$ and validation error $L_\gamma$, respectively, of GRDL with $r \in \{2, 3, 4\}, L \in \{3, 4, 5, 6\}$
.

Table 13: Number of parameters and time cost per training epoch (seconds) of GRDL and Graph transformers.

| Dataset | **Method** | | | | | |
| | GRDL | | SAT | | Graphormer | |
| | # Parameters | Time | # Parameters | Time | # Parameters | Time |
|---|---|---|---|---|---|---|
| MUTAG | **11k** | **0.11** | 663k | 0.64 | 647k | 0.59 |
| PROTEINS | **12k** | **0.52** | 666k | 3.51 | 650k | 3.31 |
| NCI1 | **13k** | **1.77** | 667k | 12.08 | 651k | 11.37 |
| IMDB-B | **15k** | **0.42** | 680k | 3.69 | 664k | 3.42 |
| IMDB-M | **13k** | **0.69** | 674k | 4.32 | 658k | 4.06 |
| PTC-MR | **12k** | **0.17** | 663k | 1.17 | 647k | 0.97 |
| BZR | **11k** | **0.21** | 665k | 1.26 | 649k | 1.09 |
| COLLAB | **31k** | **3.71** | 725k | 17.45 | 709k | 16.89 |
| Average | **15k** | **0.95** | 675k | 5.51 | 659k | 5.21 |

# E  Proof Setups

**Vector and matrix norms** The $\ell_2$-norm $\|\cdot\|_2$ is always computed entry-wise; thus, for a matrix, it corresponds to the Frobenius norm. The metric $\rho$ of function spaces is defined as the $\ell_2$-norm of the difference between the outputs of functions given some input $X$, i.e.,

$$\rho(f_1, f_2) = \|f_1(X) - f_2(X)\|_2. \tag{28}$$

Finally, let $\|\cdot\|_\sigma$ be the spectral norm and $\|\cdot\|_{p,q}$ be the $(p, q)$ matrix norm defined by $\|\mathbf{A}\|_{p,q} := \|(\|\mathbf{A}_{:,1}\|_p, \ldots, \|\mathbf{A}_{:,m}\|_p)\|_q$ for $\mathbf{A} \in \mathbb{R}^{d \times m}$.

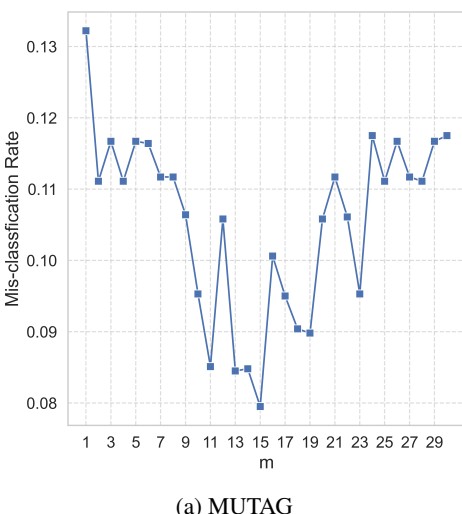

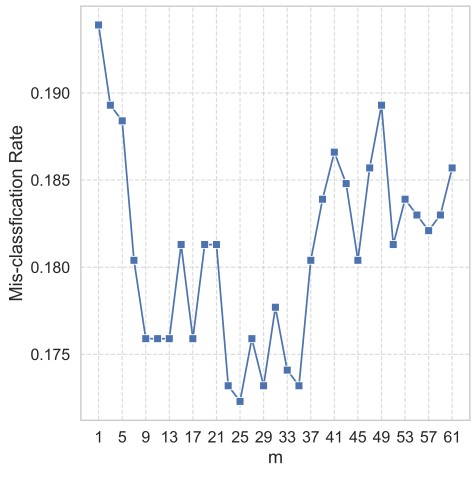

(a) MUTAG                          (b) MUTAG

Figure 6: Misclassification rate of GRDL on MUTAG (Figure 6a) and PROTEINS (Figure 6b) using different reference sizes $m$. The figures show that our model performs the best when a moderate $m$ is used.

**Rademacher complexity** Rademacher complexity is a standard complexity measure of hypothesis function space. Given dataset $\mathcal{G}$ and hypothesis function space $\mathcal{F}$, the Rademacher complexity is defined as

$$\mathcal{R}_{\mathcal{G}}(\mathcal{F}) := \underset{\sigma_1,\ldots,\sigma_N}{\mathbb{E}} \left[ \sup_{f \in \mathcal{F}} \frac{1}{N} \sum_{i=1}^{N} \sigma_i f(G_i) \right] \tag{29}$$

where $\sigma_1, \ldots, \sigma_N$ are independent Rademacher variables.

Then the bound can be derived with the help of the following lemma:

**Lemma E.1.** *Given hypothesis function space $\mathcal{F}$ that maps a graph $G \in \mathcal{X}$ to $\mathbb{R}^K$ and any $\gamma > 0$, define $\mathcal{F}_\gamma := \{(G, y) \mapsto l_\gamma(f(G), y) : f \in \mathcal{F}\}$. Then, with probability at least $1 - \delta$ over a sample $\mathcal{G}$ of size $N$, every $f \in \mathcal{F}$ satisfies $L_\gamma(f) \leq \hat{L}_\gamma(f) + 2\mathcal{R}_{\mathcal{G}}(\mathcal{F}_\gamma) + 3\gamma\sqrt{\frac{\ln(2/\delta)}{2N}}$.*

This Lemma is a standard tool in Rademacher complexity [Mohri et al., 2018]. The only problem left is to calculate the Rademacher complexity $\mathcal{R}_{\mathcal{G}}(\mathcal{F}_\gamma)$.

**Covering number complexity bounds** Direct calculation of the Rademacher complexity is often hard and the covering number is typically used to upper bound it. $V$ is an $\epsilon$-cover of $U$ with respect to some metric $\varrho$ if for all $v \in U$, there exists $v' \in V$ such that $\varrho(v, v') \leq \epsilon$, meaning

$$\sup_{v \in U} \min_{v' \in V} \varrho(v, v') \leq \epsilon. \tag{30}$$

The covering number $\mathcal{N}(\epsilon, U, \varrho)$ is defined as the least cardinality of the subset $V$. With covering number, the Rademacher complexity is upper bounded by the following Dudley entropy integral:

**Lemma E.2** (Lemma A.5 of Bartlett et al. [2017], reformulated). *Let $\mathcal{F}_\gamma$ be a real-valued function class taking values in $[0, \gamma]$, and assume that $\mathbf{0} \in \mathcal{F}_\gamma$. Then*

$$\mathcal{R}_{\mathcal{G}}(\mathcal{F}_\gamma) \leq \inf_{\alpha > 0} \left( \frac{4\alpha\gamma}{\sqrt{N}} + \frac{12}{N} \int_{\gamma\alpha}^{\gamma\sqrt{N}} \sqrt{\ln \mathcal{N}(\epsilon, \mathcal{F}_\gamma, \rho)} \, d\epsilon \right).$$

Now the only thing left is to bound $\mathcal{N}(\epsilon, \mathcal{F}_\gamma, \rho)$.

## F   Proof for Theorems, Corollaries, and Lemmas

### F.1   Correctness Analysis

We first provide the formal definition of correct classification:

**Definition F.1** (Correctness of Classification). For a graph $G_i$ with node embedding $\mathbf{H}_i$ belonging to the $k$-th class, it is correctly classified if $\mathrm{MMD}(\mathbf{H}_i, \mathbf{D}_k) < \min_{j \neq k} \mathrm{MMD}(\mathbf{H}_i, \mathbf{D}_j)$.

**Lemma F.2** (Correctness of Classification). *The classification of a graph $G_i$ belonging to the $k$-th class (with latent node embedding $\mathbf{H}_i$) is correct with probability at least $1 - \delta$ if*

$$\min_{j \neq k} \mathrm{MMD}(\mathbb{P}_k, \mathbb{P}_j) > \left(\frac{1}{\sqrt{m}} + \frac{1}{\sqrt{n}}\right)\left(4 + 4\sqrt{\log \frac{2}{\delta}}\right).$$

Lemma F.2 suggests that a larger reference distributions size ($m$) and graphs with more nodes ($n$) induce a smaller $\left(\frac{1}{\sqrt{m}} + \frac{1}{\sqrt{n}}\right)\left(4 + 4\sqrt{\log \frac{2}{\delta}}\right)$, making correct classification easier. Proof of this theorem is provided in Appendix F.10.

## F.2 Lipschitz properties

This section proves some useful lemmas related to functions' lipschitz property.

**Lemma F.3.** *For any $\mathbf{Z}, \mathbf{Z}' \in \mathbb{R}^{n \times d}$ and $\mathbf{W} \in \mathbb{R}^{d \times m}$, $\|(\mathbf{Z} - \mathbf{Z}')\mathbf{W}\|_2 \leq \|\mathbf{W}^\top\|_\sigma \|\mathbf{Z} - \mathbf{Z}'\|_2$*

*Proof.* First consider matrices $\mathbf{X}, \mathbf{Y} \in \mathbb{R}^{d \times d}$ where $\mathbf{X}, \mathbf{Y}$ are positive semi-definite. $\mathbf{Y}$ is unitarily diagonalizable, means $\mathbf{Q}\Lambda\mathbf{Q}^{-1}$ where $\Lambda = diag(\lambda_1, \dots, \lambda_d)$ is the diagonal matrix of eigenvalus of $\mathbf{Y}$. Then we have

$$\mathrm{Tr}(\mathbf{XY}) = \mathrm{Tr}(\mathbf{XQ}\Lambda\mathbf{Q}^{-1}) = \mathrm{Tr}(\Lambda\mathbf{Q}^{-1}\mathbf{XQ})$$

Let $\mathbf{P} = \mathbf{Q}^{-1}\mathbf{XQ}$, and $\lambda_0 = \max_i \lambda_i$, we have

$$\mathrm{Tr}(\mathbf{XY}) = \mathrm{Tr}(\Lambda\mathbf{P}) = \sum_k \lambda_k p_{kk} \leq \sum_k \lambda_0 p_{kk} = \lambda_0 \mathrm{Tr}(\mathbf{P}) = \lambda_0 \mathrm{Tr}(\mathbf{Q}^{-1}\mathbf{XQ}) = \lambda_0 \mathrm{Tr}(\mathbf{X})$$

Take $\mathbf{X} = (\mathbf{Z} - \mathbf{Z}')^\top(\mathbf{Z} - \mathbf{Z}')$ and $\mathbf{Y} = \mathbf{WW}^\top$, easy to see that

$$\|(\mathbf{Z}-\mathbf{Z}')\mathbf{W}\|_2 = \mathrm{Tr}\left((\mathbf{Z} - \mathbf{Z}')^\top(\mathbf{Z} - \mathbf{Z}')\mathbf{WW}^\top\right) \leq \lambda_{max}(\mathbf{WW}^\top)\mathrm{Tr}\left((\mathbf{Z} - \mathbf{Z}')^\top(\mathbf{Z} - \mathbf{Z}')\right) = \|\mathbf{W}^\top\|_\sigma\|\mathbf{Z}-\mathbf{Z}'\|_2$$

$\square$

**Lemma F.4.** *If $\sigma : \mathbb{R}^d \to \mathbb{R}^d$ is $\kappa$-Lipschitz along ever coordinate, then it is $\kappa$-Lipschitz according to $\|\cdot\|_p$ for any $p \geq 1$.*

*Proof.* For any $z, z' \in \mathbb{R}^d$,

$$\begin{aligned}
\|\sigma(z) - \sigma(z')\|_p &= \left(\sum_i |\sigma(z)_i - \sigma(z')_i|^p\right)^{1/p} \\
&\leq \left(\sum_i \kappa^p |z_i - z_i'|^p\right)^{1/p} \\
&= \kappa \|z - z'\|_p
\end{aligned}$$

$\square$

**Lemma F.5** (Lemma A.3 of Bartlett et al. [2017]). *For every $j$ and every $p \geq 1$, $\mathcal{M}(\cdot, j)$ is 2-Lipschitz w.r.t $\|\cdot\|_p$.*

*Proof.* Let $v, v', j$ be given. Without loss of generality, suppose $\mathcal{M}(v, j) \geq \mathcal{M}(v', j)$. Choose coordinate $i \neq j$ so that $\mathcal{M}(v', j) = v_j' - v_i'$. Then

$$\begin{aligned}
\mathcal{M}(v, j) - \mathcal{M}(v', j) &= (v_j - \max_{l \neq j} v_l) - (v_j' - v_i') = v_j - v_j' + v_i' + \min_{l \neq j}(-v_l) \\
&\leq (v_j - v_j') + (v_i' - v_i) \leq 2\|v - v'\|_\infty \leq 2\|v - v'\|_p
\end{aligned}$$

$\square$

**Lemma F.6.** *For every $p > 1$, $r_\zeta(-\mathcal{M}(\cdot, y))$ is $\frac{2}{\zeta}$-Lipschitz w.r.t $\|\cdot\|_p$.*

*Proof.* Recall that,

$$r_\zeta(t) := \begin{cases} 0 & t < -\zeta, \\ 1 + t/\zeta & r \in [-\zeta, 0], \\ 1 & t > 0, \end{cases} \tag{31}$$

Then the proof is trivial be Lemma F.5. □

**Lemma F.7.** *Cross entropy loss $l(\mathbf{x}, \mathbf{y}) = -\sum_{k=1}^{K} y_k \log \frac{\exp(x_k)}{\sum_{j=1}^{K} \exp(x_j)}$ is $\sqrt{2}$-Lipschitz w.r.t $\|\cdot\|_2$.*

*Proof.* Since $l$ is differentiable, it is sufficient to find $\mu$ such that $\|\nabla l\|_2 \leq \mu$. Without loss of generality, suppose $y_i = 1$, then $l(\mathbf{x}, \mathbf{y}) = -\log \frac{\exp(x_i)}{\sum_{j=1}^{K} \exp(x_j)}$. Let $s = \sum_k \exp x_k$, we have

$$\nabla l_i = -\frac{\sum_{l \neq i} \exp(x_l)}{s}, \qquad \nabla l_j = \frac{\exp(x_j)}{s} \quad \forall j \neq i$$

Therefore

$$\|\nabla l\|_2^2 = \frac{(\sum_{l \neq i} \exp(x_l))^2 + \sum_{l \neq i} \exp(2x_l)}{s^2} \overset{(a)}{\leq} 2 \frac{(\sum_{l \neq i} \exp(x_l))^2}{s^2} \leq 2$$

where $(a)$ is because Cauchy–Schwarz inequality. □

## F.3 Covering number

The following lemma provides an upper bound of the covering number for the network $\mathcal{F}_G$.

**Lemma F.8** (Covering number bound of $\mathcal{F}_G$). *Let $c = \|\tilde{\mathbf{A}}\|_\sigma$ and $\bar{d} = \max_{i,l} d_i^{(l)}$. Given an $L$-layer GIN message passing network $\mathcal{F}_G$, for any $\epsilon > 0$*

$$\ln \mathcal{N}(\epsilon, \mathcal{F}_G, \rho) \leq \frac{R_G}{\epsilon^2}$$

*where $R_G = c^{2L} \|\mathbf{X}\|_2^2 \ln(2\bar{d}^2) \left( \prod_{l=1}^{L} \kappa_l^2 \right) \left( \sum_{l=1}^{L} (\tau_l)^{\frac{2}{3}} \right)^3$ and $\kappa_l = \prod_{j=1}^{r} \kappa_j^{(l)}$, $\tau_l = \left( \sum_{i=1}^{r} \left( \frac{b_i^{(l)}}{\kappa_i^{(l)}} \right)^{2/3} \right)^{3/2}$.*

Firstly, we introduce the core lemma used to find the covering number of compositions of multiple hypothesis function classes.

**Lemma F.9.** *Given hypothesis function classes $\mathcal{F}_1, \mathcal{F}_2, \ldots \mathcal{F}_k$ that maps input from matrix space to output in matrix space and their covering radius $(\epsilon_1, \epsilon_2, \ldots, \epsilon_k)$. Assume all functions in $\mathcal{F}_i$ is $\kappa_i$-Lipschitz w.r.t. $\|\cdot\|_2$, and $\ln \mathcal{N}(\epsilon_i, \mathcal{F}_i, \rho) \leq g(\eta_i)$ for some function $g$ with parameters $\eta_i$ ($\eta_i$ can be multi-valued). Then there exists $\epsilon$-cover $\mathcal{C}$ of $\mathcal{F} = \mathcal{F}_k \circ \mathcal{F}_{k-1} \circ \cdots \mathcal{F}_1$ with $\epsilon = \sum_{i=1}^{k} \left( \epsilon_i \prod_{j=i+1}^{k} \kappa_j \right)$ such that*

$$\ln |\mathcal{C}| \leq \sum_{i=1}^{k} g(\eta_i)$$

*Proof.* Inductively construct covers as follows.

- Let $\mathcal{C}_1$ be the $\epsilon_1$-cover of $\mathcal{F}_1$. By our assumption,

$$\ln |\mathcal{C}_1| \leq g(\eta_1)$$

- Let $\mathcal{C}_j$ as a $\epsilon_j$-cover of $\mathcal{F}_j \circ \cdots \circ \mathcal{F}_1$. Suppose $\ln |\mathcal{C}_j| \leq \sum_{i=1}^{j} g(\eta_i)$. For every $f'_j \circ \ldots f'_1 \in \mathcal{C}_j$, we construct $\mathcal{C}_{j+1, f'_j, \ldots, f'_1}$ as an $\epsilon_{j+1}$-cover of $\mathcal{F}_{j+1} \circ f'_j \circ \cdots \circ f'_1$. Define

$$\mathcal{C}_{j+1} := \bigcup_{f'_h \in \mathcal{C}_h, h \leq j} \mathcal{C}_{j+1, f'_j, \ldots, f'_1}$$

It is clearly a cover of $\mathcal{F}_{j+1} \circ \mathcal{F}_j \circ \cdots \circ \mathcal{F}_1$ By our assumption, we know that

$$\ln |\mathcal{C}_{j+1,f'_j,\ldots,f'_1}| \leq g(\eta_{j+1})$$
$$|\mathcal{C}_{j+1,f'_j,\ldots,f'_1}| \leq \exp\left(g(\eta_{j+1})\right)$$

Then, it is trivial to see

$$|\mathcal{C}_{j+1}| \leq |\mathcal{C}_j| \exp\left(g(\eta_j)\right)$$
$$\ln |\mathcal{C}_{j+1}| \leq \sum_{i=1}^{j+1} g(\eta_i)$$

By the inductive arguments above, we can conclude that

$$\ln |\mathcal{C}| = \ln |\mathcal{C}_k| \leq \sum_{i=1}^{k} g(\eta_i)$$

Next, inductively find the cover radius $\epsilon$ for $\mathcal{C}$.

- It is trivial in the base case that the cover of $\mathcal{C}_1$ is $\epsilon_1$.
- Suppose for $\mathcal{C}_h$, the cover radius satisfies

$$\epsilon_h = \sum_{i=1}^{h} \left( \epsilon_i \prod_{j=i+1}^{h} \kappa_j \right)$$

For all $f_{h+1} \circ f_h \circ \cdots \circ f_1 \in \mathcal{F}_{h+1} \circ \mathcal{F}_h \circ \cdots \circ \mathcal{F}_1$, there exists $f'_{h+1} \circ f'_h \circ \cdots \circ f'_1 \in \mathcal{C}_{h+1}$ such that

$$
\begin{aligned}
\rho(f'_{h+1} \circ f'_h \circ \cdots \circ f'_1, f_{h+1} \circ f_h \circ \cdots \circ f_1) &\leq \rho(f'_{h+1} \circ f'_h \circ \cdots \circ f'_1, f_{h+1} \circ f'_h \circ \cdots \circ f'_1) \\
&\quad + \rho(f_{h+1} \circ f'_h \circ \cdots \circ f'_1, f_{h+1} \circ f_h \circ \cdots \circ f_1) \\
&\leq \epsilon_{h+1} + \kappa_{h+1} \rho(f'_h \circ \cdots \circ f'_1, f_h \circ \cdots \circ f_1) \\
&\leq \epsilon_{h+1} + \kappa_{h+1} \epsilon_h \\
&= \sum_{i=1}^{h+1} \left( \epsilon_i \prod_{j=i+1}^{h+1} \kappa_j \right)
\end{aligned}
$$

By the inductive arguments,

$$\epsilon = \epsilon_k = \sum_{i=1}^{k} \left( \epsilon_i \prod_{j=i+1}^{k} \kappa_j \right)$$

$\square$

The following matrix covering number is well-known and the detailed proof can be found in Bartlett et al. [2017].

**Lemma F.10.** *Let conjugate exponents $(p, q)$ and $(r, s)$ be given with $p \leq 2$, as well as positive reals $(a, b, \epsilon)$ and positive integer m. Let matrix $\mathbf{X} \in \mathbb{R}^{n \times d}$ be given with $\|\mathbf{X}\|_p \leq b$. Then*

$$\ln \mathcal{N}\left(\left\{\mathbf{XA} : \mathbf{A} \in \mathbb{R}^{d \times m}, \|\mathbf{A}\|_{q,s} \leq a\right\}, \epsilon, \|\cdot\|_2\right) \leq \left\lceil \frac{a^2 b^2 m^{2/r}}{\epsilon^2} \right\rceil \ln(2dm)$$

For the composition of a hypothesis function class and a $\kappa$-Lipschitz function, we have the following lemma

**Lemma F.11.** *Suppose $\psi$ is a $\kappa$-Lipschitz function, then $\ln \mathcal{N}(\epsilon, \psi \circ \mathcal{F}, \rho) \leq \ln \mathcal{N}(\epsilon/\kappa, \mathcal{F}, \rho)$*

*Proof.* Let $\mathcal{C}$ denote $\frac{\epsilon}{\kappa}$-cover of $\mathcal{F}$, then for any $f \in \mathcal{F}$, there exists $f' \in \mathcal{C}$ such that $\rho(f, f') \leq \frac{\epsilon}{\kappa}$. Let $\mathbf{Z}$ denote the input, we have

$$\rho(\psi \circ f', \psi \circ f) = \|\psi \circ f'(\mathbf{Z}) - \psi \circ f(\mathbf{Z})\|_2$$
$$\leq \kappa \|f'(\mathbf{Z}) - f(\mathbf{Z}zh)\|_2 \qquad \text{(Lemma F.4)}$$
$$= \kappa \rho(f', f)$$
$$\leq \epsilon$$

$\square$

**Lemma F.12.** *Given* $\mathbf{H}, \mathbf{H}' \in \mathbb{R}^{n \times d}$ *and* $\mathbf{D}, \mathbf{D}' \in \mathbb{R}^{m \times d}$. *Squared MMD distance with gaussian kernel* $k(x, y) = \exp\left\{-\theta \|x - y\|_2^2\right\}$ *satisfies*

$$\left|\text{MMD}^2\left(\mathbf{H}, \mathbf{D}\right) - \text{MMD}^2\left(\mathbf{H}', \mathbf{D}'\right)\right| \leq 4\sqrt{\theta}\left(n^{-1/2}\|\mathbf{H} - \mathbf{H}'\|_2 + m^{-1/2}\|\mathbf{D} - \mathbf{D}'\|_2\right)$$

*Proof.* The matrices have the form

$$\mathbf{H} = \begin{bmatrix} \mathbf{h}_1^\top \\ \mathbf{h}_2^\top \\ \vdots \\ \mathbf{h}_n^\top \end{bmatrix} \qquad \mathbf{H}' = \begin{bmatrix} \mathbf{h'}_1^\top \\ \mathbf{h'}_2^\top \\ \vdots \\ \mathbf{h'}_n^\top \end{bmatrix} \qquad \mathbf{D} = \begin{bmatrix} \mathbf{d}_1^\top \\ \mathbf{d}_2^\top \\ \vdots \\ \mathbf{d}_m^\top \end{bmatrix} \qquad \mathbf{D}' = \begin{bmatrix} \mathbf{d'}_1^\top \\ \mathbf{d'}_2^\top \\ \vdots \\ \mathbf{d'}_m^\top \end{bmatrix}$$

Then we have

$$\left|\text{MMD}^2\left(\mathbf{H}, \mathbf{D}\right) - \text{MMD}^2\left(\mathbf{H}', \mathbf{D}'\right)\right| \leq \left| \frac{1}{n^2} \sum_{i,j=1}^{n} \left[\exp\left(-\theta\|\mathbf{h}_i - \mathbf{h}_j\|_2^2\right) - \exp\left(-\theta\|\mathbf{h}'_i - \mathbf{h}'_j\|_2^2\right)\right]\right|$$

$$+ \left|\frac{1}{m^2} \sum_{i,j=1}^{m} \left[\exp\left(-\theta\|\mathbf{d}_i - \mathbf{d}_j\|_2^2\right) - \exp\left(-\theta\|\mathbf{d}'_i - \mathbf{d}'_j\|_2^2\right)\right]\right|$$

$$+ \left|\frac{2}{mn} \sum_{i=1}^{n}\sum_{j=1}^{m} \left[\exp\left(-\theta\|\mathbf{h}_i - \mathbf{d}_j\|_2^2\right) - \exp\left(-\theta\|\mathbf{h}'_i - \mathbf{d}'_j\|_2^2\right)\right]\right|$$

$$\overset{(a)}{\leq} \frac{\sqrt{\theta}}{n^2} \sum_{i,j=1}^{n} \left|\|\mathbf{h}_i - \mathbf{h}_j\|_2 - \|\mathbf{h}'_i - \mathbf{h}'_j\|_2\right| + \frac{\sqrt{\theta}}{m^2} \sum_{i,j=1}^{m} \left|\|\mathbf{d}_i - \mathbf{d}_j\|_2 - \|\mathbf{d}'_i - \mathbf{d}'_j\|_2\right|$$

$$+ \frac{2\sqrt{\theta}}{mn} \sum_{i=1}^{n}\sum_{j=1}^{m} \left|\|\mathbf{h}_i - \mathbf{d}_j\|_2 - \|\mathbf{h}'_i - \mathbf{d}'_j\|_2\right|$$

$$\overset{(b)}{\leq} \frac{\sqrt{\theta}}{n^2} \sum_{i,j=1}^{n} \|(\mathbf{h}_i - \mathbf{h}'_i) - (\mathbf{h}_j - \mathbf{h}'_j)\|_2 + \frac{\sqrt{\theta}}{m^2} \sum_{i,j=1}^{m} \|(\mathbf{d}_i - \mathbf{d}'_i) - (\mathbf{d}_j - \mathbf{d}'_j)\|_2$$

$$+ \frac{2\sqrt{\theta}}{mn} \sum_{i=1}^{n}\sum_{j=1}^{m} \|(\mathbf{h}_i - \mathbf{h}'_i) - (\mathbf{d}_j - \mathbf{d}'_j)\|_2$$

$$\leq \frac{4\sqrt{\theta}}{n} \sum_{i=1}^{n} \|\mathbf{h}_i - \mathbf{h}'_i\|_2 + \frac{4\sqrt{\theta}}{m} \sum_{j=1}^{m} \|\mathbf{d}_i - \mathbf{d}'_i\|_2$$

$$\overset{(c)}{\leq} \frac{4\sqrt{\theta n}}{n}\|\mathbf{H} - \mathbf{H}'\|_2 + \frac{4\sqrt{\theta m}}{m}\|\mathbf{D} - \mathbf{D}'\|_2$$

$$= 4\sqrt{\theta}\left(n^{-1/2}\|\mathbf{H} - \mathbf{H}'\|_2 + m^{-1/2}\|\mathbf{D} - \mathbf{D}'\|_2\right).$$

In the above derivation, (a) holds due to $\left|\exp\left(-x^2\right) - \exp\left(-y^2\right)\right| \leq |x - y|$ for any $x, y \geq 0$, (b) holds due to the triangle inequality, and (c) holds by the Cauchy–Schwarz inequality. $\square$

The covering number for a single GIN message passing layer $\mathcal{F}^l$ with the following Lemma:

**Lemma F.13** (Covering number of $\mathcal{F}^l$). *Let $c = \|\tilde{\mathbf{A}}\|_\sigma$. For any $l \in [L]$ and $\epsilon > 0$*

$$\ln \mathcal{N}\left(\epsilon, \mathcal{F}^l, \rho\right) \leq \frac{c^{2l} \tau_l^2 \left(\prod_{i=1}^l \kappa_i\right)^2}{\epsilon^2} \|\mathbf{X}\|_2^2 \ln(2\bar{d}^2),$$

*where $\kappa_i = \prod_{j \leq r} \kappa_j^{(i)}$, $\tau_l = \left(\sum_{i=1}^r \left(\frac{b_i^{(l)}}{\kappa_i^{(l)}}\right)^{2/3}\right)^{3/2}$.*

*Proof.* With a little abuse of notation, remove the superscript in Equation (19) for now

$$\mathcal{F}^l = \{(\tilde{\mathbf{A}}, \mathbf{H}) \mapsto \sigma\left(\cdots \sigma\left(\left(\tilde{\mathbf{A}}\mathbf{H}\right)\mathbf{W}_1\right)\cdots \mathbf{W}_{r-1}\right)\mathbf{W}_r : \mathbf{W}_i \in \mathcal{B}_i\}$$

where $\mathcal{B}_i := \left\{\mathbf{W}_i : \|\mathbf{W}_i^\top\|_\sigma \leq \kappa_i, \|\mathbf{W}_i\|_{2,1} \leq b_i\right\}$. Denote $\mathcal{F}_i = \{\mathbf{Z} \mapsto \sigma(\mathbf{Z}\mathbf{W_i}) : \mathbf{W}_i \in \mathcal{B}_i\}$ for $i \in [r-1]$, $\mathcal{F}_r = \{\mathbf{Z} \mapsto \mathbf{Z}\mathbf{W_r} : \mathbf{W}_r \in \mathcal{B}_r\}$, then

$$\mathcal{F}^l = \mathcal{F}_r \circ \mathcal{F}_{r-1} \circ \cdots \mathcal{F}_1$$

For any $f_i \in \mathcal{F}_i, i \in [r-1]$ with arbitrary input $\mathbf{Z}, \mathbf{Z}'$

$$
\begin{aligned}
\|f_i(\mathbf{Z}) - f_i(\mathbf{Z}')\|_2 &= \|\sigma(\mathbf{Z}\mathbf{W}_i) - \sigma(\mathbf{Z}'\mathbf{W}_i)\|_2 \\
&\leq \|\mathbf{Z}\mathbf{W}_i - \mathbf{Z}'\mathbf{W}_i\|_2 && \text{(Lemma F.4)} \\
&\leq \|\mathbf{W}_i^\top\|_\sigma \|\mathbf{Z} - \mathbf{Z}'\|_2 && \text{(Lemma F.3)} \\
&= \kappa_i \|\mathbf{Z} - \mathbf{Z}'\|_2
\end{aligned}
$$

Similarly, for any $f_r \in \mathcal{F}_r'$, Lemma F.3 gives

$$\|f_r(\mathbf{Z}) - f_r(\mathbf{Z}')\|_2 = \|\mathbf{Z}\mathbf{W}_r - \mathbf{Z}'\mathbf{W}_r\|_2 \leq \kappa_r \|\mathbf{Z} - \mathbf{Z}'\|_2$$

Denoting $\mathbf{Z}_{i-1}$ as the input to $\mathcal{F}_i$ ($\mathbf{Z}_0 = \mathbf{Z} = \tilde{\mathbf{A}}\mathbf{H}$) and using the Lipschitz conditions, we have

$$f_i(f_{i-1}(\ldots f_1(\mathbf{Z}))) \leq \left(\prod_{j=1}^i \kappa_j\right) \|\mathbf{Z}\|_2 \leq c \left(\prod_{j=1}^i \kappa_j\right) \|\mathbf{H}\|_2 \qquad (32)$$

for any $f_i \circ f_{i-1} \circ \cdots \circ f_1 \in \mathcal{F}_i \circ \mathcal{F}_{i-1} \circ \cdots \circ \mathcal{F}_1$. So $\|\mathbf{Z}_{i-1}\|_2 \leq c \left(\prod_{j=1}^{i-1} \kappa_j\right) \|\mathbf{H}\|_2 \triangleq c_{i-1}$. By Lemma F.10 and Lemma F.11, we have

$$\ln \mathcal{N}\left(\epsilon_i, \mathcal{F}_i, \rho\right) \leq \frac{b_i^2 c_{i-1}^2}{\epsilon_i^2} \ln(2\bar{d}^2)$$

$W$ is the maximum dimension of weight matrices (as previously defined in the main text). Thus by Lemma F.9, we have the covering number

$$\ln \mathcal{N}\left(\epsilon, \mathcal{F}^l, \rho\right) \leq \sum_{i=1}^r \frac{b_i^2 c_{i-1}^2}{\epsilon_i^2} \ln(2\bar{d}^2) = c^2 \|\mathbf{H}\|_2^2 \ln(2\bar{d}^2) \sum_{i=1}^r \frac{b_i^2 \left(\prod_j^{i-1} \kappa_j\right)^2}{\epsilon_i^2}$$

with cover radius $\epsilon = \sum_{i=1}^r \left(\epsilon_i \prod_{j=i+1}^r \kappa_j\right)$. Next we need to choose $\epsilon_i$ to minimize the right hand side of the above inequality. Holder's inequality states that when $\frac{1}{p} + \frac{1}{q} = 1$,

$$\langle \mathbf{a}, \mathbf{b}\rangle \leq \|\mathbf{a}\|_p \|\mathbf{b}\|_q$$

$$\sum_i a_i b_i \leq \left(\sum_i a_i^p\right)^{1/p} \left(\sum_i b_i^q\right)^{1/q}$$

Let $\alpha_i^2 = b_i^2 \left( \prod_j^{i-1} \kappa_j \right)^2, \beta_i = \prod_{j=i+1}^r \kappa_j$. Choose $p = \frac{1}{3}, q = \frac{2}{3}$,

$$\left[ \sum_i \left( \frac{\alpha_i}{\epsilon_i} \right)^{\frac{2}{3} \times 3} \right]^{\frac{1}{3}} \left[ \sum_i \left( \beta_i \epsilon_i \right)^{\frac{2}{3} \times \frac{3}{2}} \right]^{\frac{2}{3}} \geq \sum_i \left( \alpha_i \beta_i \right)^{\frac{2}{3}}$$

$$\left( \sum_i \left( \frac{\alpha_i}{\epsilon_i} \right)^2 \right) \left( \sum_i \beta_i \epsilon_i \right)^2 \geq \left( \sum_i \left( \alpha_i \beta_i \right)^{\frac{2}{3}} \right)^3$$

$$\left( \sum_{i=1}^r \frac{b_i^2 \left( \prod_{j=1}^{i-1} \kappa_j \right)^2}{\epsilon_i^2} \right) \left( \sum_{i=1}^r \left( \epsilon_i \prod_{j=i+1}^r \kappa_j \right) \right)^2 \geq \prod_{j=1}^r \kappa_j^2 \left( \sum_{i=1}^r \left( \frac{b_i}{\kappa_i} \right)^{\frac{2}{3}} \right)^3 \tag{33}$$

$$\sum_{i=1}^r \frac{b_i^2 \left( \prod_{j=1}^{i-1} \kappa_j \right)^2}{\epsilon_i^2} \geq \frac{1}{\epsilon^2} \prod_{j=1}^r \kappa_j^2 \left( \sum_{i=1}^r \left( \frac{b_i}{\kappa_i} \right)^{\frac{2}{3}} \right)^3$$

Add the superscript back,

$$\ln \mathcal{N} \left( \epsilon, \mathcal{F}^l, \rho \right) \leq \ln(2\bar{d}^2) \frac{c^2 \|\mathbf{H}^{(l-1)}\|_2^2}{\epsilon^2} \prod_{j=1}^r \left( \kappa_j^{(l)} \right)^2 \left( \sum_{i=1}^r \left( \frac{b_i^{(l)}}{\kappa_i^{(l)}} \right)^{\frac{2}{3}} \right)^3$$

$\mathbf{H}^{(l-1)} = f^{(l-1)} \left( \tilde{\mathbf{A}} f^{(l-2)} \left( \dots f^{(1)} \left( \tilde{\mathbf{A}} \mathbf{X} \right) \right) \right)$ where $f^{(k)} \in \mathcal{F}^k$ for $k \in [l-1]$. By Equation (32), it is easy to see

$$\|\mathbf{H}^{(l-1)}\|_2 \leq c \prod_{j=1}^r \kappa_j^{(l-1)} \|f^{(l-2)} \left( \dots f^{(1)} \left( \tilde{\mathbf{A}} \mathbf{X} \right) \right) \|_2$$

$$\leq \dots$$

$$\leq c^{l-1} \|\mathbf{X}\|_2 \prod_{i=1}^{l-1} \left( \prod_{j=1}^r \kappa_j^{(i)} \right)$$

$$= c^{l-1} \|\mathbf{X}\|_2 \prod_{i \leq l-1, j \leq r} \kappa_j^{(i)}$$

Letting $\kappa_i = \prod_{j \leq r} \kappa_j^{(i)}, \tau_l = \left( \sum_{i=1}^r \left( \frac{b_i^{(l)}}{\kappa_i^{(l)}} \right)^{2/3} \right)^{3/2}$, we finish the proof, i.e.,

$$\ln \mathcal{N} \left( \epsilon, \mathcal{F}^l, \rho \right) \leq \frac{c^{2l} \tau_l^2 \left( \prod_{i=1}^l \kappa_i \right)^2}{\epsilon^2} \|\mathbf{X}\|_2^2 \ln(2\bar{d}^2)$$

$\square$

With the covering number of $\mathcal{F}_G$, we can calculate the covering number of $\mathcal{F}$ in Equation (10).

**Lemma F.14** (Covering number of $\mathcal{F}$). *Suppose $\theta$ in the kernel (Equation (15)) is fixed. For any $\epsilon > 0$*

$$\ln \mathcal{N} \left( \epsilon, \mathcal{F}, \rho \right) \leq \frac{64 \theta K R_G}{n \epsilon^2} + K m d \ln \left( \frac{24 b_D \sqrt{\theta N}}{\sqrt{m} \epsilon} \right)$$

*where $R_G$ is defined the same as Lemma F.8.*

*Proof.* Denote $\mathbf{S} \in \mathbb{R}^{N \times K}$ as the output of function $\mathcal{F}$. Consider the entry $(i, j)$ of $\mathbf{S}$, by Lemma F.12, it has

$$|s_{ij} - s'_{ij}| = \left| \text{MMD}^2 \left( \mathbf{H}_i, \mathbf{D}_j \right) - \text{MMD}^2 \left( \mathbf{H}'_i, \mathbf{D}'_j \right) \right|$$

$$\leq 4\sqrt{\theta} \left( n^{-1/2} \|\mathbf{H}_i - \mathbf{H}'_i\|_2 + m^{-1/2} \|\mathbf{D}_j - \mathbf{D}'_j\|_2 \right)$$

Then for the whole matrix $\mathbf{S}$,

$$
\begin{aligned}
\|\mathbf{S} - \mathbf{S}'\|_2 &= \sqrt{\sum_{i=1}^{N} \sum_{j=1}^{K} |s_{ij} - s'_{ij}|^2} \\
&\leq 4\sqrt{\theta} \sqrt{\sum_{i=1}^{N} \sum_{j=1}^{K} \left(n^{-1/2}\|\mathbf{H}_i - \mathbf{H}'_i\|_2 + m^{-1/2}\|\mathbf{D}_j - \mathbf{D}'_j\|_2\right)^2} \\
&\stackrel{(a)}{\leq} 4\sqrt{\theta} \sqrt{\sum_{i=1}^{N} \sum_{j=1}^{K} 2\left(n^{-1}\|\mathbf{H}_i - \mathbf{H}'_i\|_2^2 + m^{-1}\|\mathbf{D}_j - \mathbf{D}'_j\|_2^2\right)} \\
&= 4\sqrt{2\theta} \sqrt{\left(Kn^{-1}\|\mathbf{H} - \mathbf{H}'\|_2^2 + Nm^{-1}\|\mathbf{D} - \mathbf{D}'\|_2^2\right)}
\end{aligned}
$$

where (a) holds due to $(x + y)^2 \leq 2(x^2 + y^2)$. If $\|\mathbf{H} - \mathbf{H}'\|_2 \leq \epsilon_1$ and $\|\mathbf{D} - \mathbf{D}'\|_2 \leq \epsilon_2$, then

$$
\begin{aligned}
\|\mathbf{S} - \mathbf{S}'\|_2 &\leq 4\sqrt{2\theta} \sqrt{\left(Kn^{-1}\epsilon_1^2 + Nm^{-1}\epsilon_2^2\right)} \\
&\leq \epsilon
\end{aligned}
$$

by choosing $\epsilon_1 = \frac{\sqrt{n}}{8\sqrt{K\theta}}\epsilon$ and $\epsilon_2 = \frac{\sqrt{m}}{8\sqrt{N\theta}}\epsilon$. Let $\mathcal{B}_D := \{\mathbf{D} \in \mathbb{R}^{Km \times d} : \|\mathbf{D}\|_2 \leq b_D\}$. It is well-known that there exists an $\epsilon_2$-cover obeying

$$
\mathcal{N}\left(\epsilon_2, \mathcal{B}_D, \|\cdot\|_2\right) \leq \left(\frac{3b_D}{\epsilon_2}\right)^{Kmd} \tag{34}
$$

Denote the output space of $\mathcal{F}_G, \mathcal{F}$ as $\mathcal{H}, \mathcal{Z}$ respectively, we can bound the covering number as

$$
\begin{aligned}
\mathcal{N}\left(\epsilon, \mathcal{F}, \rho\right) &= \mathcal{N}\left(\epsilon, \mathcal{Z}, \|\cdot\|_2\right) \\
&\leq \mathcal{N}\left(\epsilon_1, \mathcal{H}, \|\cdot\|_2\right) \mathcal{N}\left(\epsilon_2, \mathcal{B}_D, \|\cdot\|_2\right)
\end{aligned}
$$

It follows from Lemma F.8 and inequality (34) that

$$
\begin{aligned}
\ln \mathcal{N}\left(\epsilon, \mathcal{F}, \rho\right) &\leq \ln \mathcal{N}\left(\epsilon_1, \mathcal{H}, \|\cdot\|_2\right) + \ln \mathcal{N}\left(\epsilon_2, \mathcal{B}_D, \|\cdot\|_2\right) \\
&\leq \frac{64\theta K R_G}{n\epsilon^2} + Kmd \ln\left(\frac{24b_D\sqrt{\theta N}}{\sqrt{m}\epsilon}\right)
\end{aligned}
$$

$\square$

## F.4 Proof of Lemma E.2

The Dudley entropy integral bound used by Bartlett et al. [2017] is

**Lemma F.15** (Lemma A.5 of Bartlett et al. [2017]). *Let $\mathcal{F}$ be a real-valued function class taking values in $[0, 1]$, and assume that $\mathbf{0} \in \mathcal{F}$. Then*

$$
\mathcal{R}_{\mathcal{G}}(\mathcal{F}) \leq \inf_{\alpha > 0} \left(\frac{4\alpha}{\sqrt{N}} + \frac{12}{N} \int_{\alpha}^{\sqrt{N}} \sqrt{\ln \mathcal{N}\left(\epsilon, \mathcal{F}, \rho\right)} \, d\epsilon\right).
$$

Lemma E.2 can be proved with simple modifications.

*Proof.* Let $\mathcal{F}' = \psi \circ \mathcal{F}_\gamma$ where $\psi(x) = \frac{1}{\gamma}x$, then $\mathcal{F}'$ is a real-valued function class taking values in $[0, 1]$. By Lemma F.15, it has

$$\mathcal{R}_\mathcal{G}(\mathcal{F}') \leq \inf_{\alpha>0} \left( \frac{4\alpha}{\sqrt{N}} + \frac{12}{N} \int_\alpha^{\sqrt{N}} \sqrt{\ln\mathcal{N}\left(\frac{\epsilon}{\gamma}, \mathcal{F}', \rho\right)} \, d(\frac{\epsilon}{\gamma}) \right)$$

$$= \inf_{\alpha>0} \left( \frac{4\alpha}{\sqrt{N}} + \frac{12}{N\gamma} \int_{\gamma\alpha}^{\gamma\sqrt{N}} \sqrt{\ln\mathcal{N}\left(\frac{\epsilon}{\gamma}, \psi \circ \mathcal{F}_\gamma, \rho\right)} \, d\epsilon \right)$$

$$\leq \inf_{\alpha>0} \left( \frac{4\alpha}{\sqrt{N}} + \frac{12}{N\gamma} \int_{\gamma\alpha}^{\gamma\sqrt{N}} \sqrt{\ln\mathcal{N}(\epsilon, \mathcal{F}_\gamma, \rho)} \, d\epsilon \right) \qquad \text{(Lemma F.11)}$$

Multiplying both side by $\gamma$, it has

$$\mathcal{R}_\mathcal{G}(\mathcal{F}_\gamma) \leq \inf_{\alpha>0} \left( \frac{4\alpha\gamma}{\sqrt{N}} + \frac{12}{N} \int_{\gamma\alpha}^{\gamma\sqrt{N}} \sqrt{\ln\mathcal{N}(\epsilon, \mathcal{F}_\gamma, \rho)} \, d\epsilon \right).$$

$\square$

## F.5 Proof of Lemma F.8

*Proof.* By Lemma F.13, we know

$$\ln\mathcal{N}\left(\epsilon_l, \mathcal{F}^l, \rho\right) \leq \frac{c^{2l}\tau_l^2 \left(\prod_{i=1}^l \kappa_i\right)^2}{\epsilon_l^2} \|\mathbf{X}\|_2^2 \ln(2\bar{d}^2)$$

For any $f_l \in \mathcal{F}^l$, given input $\mathbf{Z}$, it has

$$\|f_l(\mathbf{Z})\|_2 = \|\sigma\left(\cdots\sigma\left(\left(\tilde{\mathbf{A}}\mathbf{Z}\right)\mathbf{W}_1^{(l)}\right)\cdots\mathbf{W}_{r-1}^{(l)}\right)\mathbf{W}_r^{(l)}\|_2$$

$$\leq \kappa_r^{(l)} \|\sigma\left(\cdots\sigma\left(\left(\tilde{\mathbf{A}}\mathbf{Z}\right)\mathbf{W}_1^{(l)}\right)\cdots\mathbf{W}_{r-1}^{(l)}\right)\|_2 \qquad \text{(Lemma F.3)}$$

$$\leq \|\sigma\left(\cdots\sigma\left(\left(\tilde{\mathbf{A}}\mathbf{Z}\right)\mathbf{W}_1^{(l)}\right)\cdots\mathbf{W}_{r-2}^{(l)}\right)\mathbf{W}_{r-1}^{(l)}\|_2 \qquad \text{(Lemma F.4)}$$

$$\leq \cdots$$

$$\leq \left(\prod_{j=1}^r \kappa_j^{(l)}\right) \|\mathbf{A}\mathbf{Z}\|_2$$

$$\leq c \left(\prod_{j=1}^r \kappa_j^{(l)}\right) \|\mathbf{Z}\|_2$$

$$= c\kappa_l \|\mathbf{Z}\|_2$$

which means all $f_l \in \mathcal{F}^l$ is $c\kappa_l$-Lipschitz. Applying Lemma F.9, we have

$$\ln\mathcal{N}(\epsilon, \mathcal{F}', \rho) \leq \sum_{l=1}^L \frac{\tau_l^2 \left(\prod_{j=1}^l c\kappa_j\right)^2}{\epsilon_l^2} \|\mathbf{X}\|_2^2 \ln(2\bar{d}^2)$$

with $\epsilon = \sum_{l=1}^L \left(\epsilon_l \prod_{j=l+1}^r c\kappa_j\right)$. The only thing left is to minimize $\sum_{l=1}^L \frac{\tau_l^2(\prod_{j=1}^l c\kappa_j)^2}{\epsilon_l^2}$ by controlling $\epsilon_l$'s. Choose $\alpha_l^2 = \tau_l^2 \left(\prod_{j=1}^l c\kappa_j\right)^2$, $\beta_l = \prod_{j=l+1}^L c\kappa_j$. Choose $p = \frac{1}{3}, q = \frac{2}{3}$, and apply

Holder's inequality in the same ways as Equation (33), this yields

$$\left(\sum_l \left(\frac{\alpha_l}{\epsilon_l}\right)^2\right)\left(\sum_l \beta_l \epsilon_l\right)^2 \geq \left(\sum_l (\alpha_l \beta_l)^{\frac{2}{3}}\right)^3$$

$$\left(\sum_{l=1}^{L} \frac{\tau_l^2 \left(\prod_{j=1}^{l} c\kappa_j\right)^2}{\epsilon_l^2}\right)\left(\sum_{l=1}^{L} \left(\epsilon_l \prod_{j=l+1}^{L} c\kappa_j\right)\right)^2 \geq c^{2L} \prod_{j=1}^{L} \kappa_j^2 \left(\sum_{l=1}^{L} (\tau_l)^{\frac{2}{3}}\right)^3$$

$$\sum_{l=1}^{L} \frac{\tau_l^2 \left(\prod_{j=1}^{l} c\kappa_j\right)^2}{\epsilon_l^2} \geq \frac{1}{\epsilon^2} c^{2L} \prod_{j=1}^{L} \kappa_j^2 \left(\sum_{l=1}^{L} (\tau_l)^{\frac{2}{3}}\right)^3$$

Thus derives the conclusion

$$\ln \mathcal{N}(\epsilon, \mathcal{F}_G, \rho) \leq \frac{R_G}{\epsilon^2}$$

where $R_G = c^{2L} \|\mathbf{X}\|_2^2 \ln(2\bar{d}^2) \left(\prod_{l=1}^{L} \kappa_l^2\right) \left(\sum_{l=1}^{L} (\tau_l)^{\frac{2}{3}}\right)^3$. $\qquad\square$

## F.6   Proof of Theorem 3.2

*Proof.* Since $l_\gamma(\cdot, y)$ is $\mu$-Lipschitz, we can bound the covering number of $\mathcal{F}_\gamma$ (defined in Lemma E.1)

$$\ln \mathcal{N}(\epsilon, \mathcal{F}_\gamma, \rho) \leq \ln \mathcal{N}\left(\frac{\epsilon}{\mu}, \mathcal{F}, \rho\right) \qquad \text{(Lemma F.11)}$$

$$\leq \frac{64\theta K R_G \mu^2}{n\epsilon^2} + Kmd \ln\left(\frac{24 b_D \mu \sqrt{\theta N}}{\sqrt{m}\epsilon}\right) \qquad \text{(Lemma F.14)}$$

Denote $v_1 = \frac{64\theta K R_G \mu^2}{n}, v_2 = Km\bar{d}, v_3 = \frac{24\sqrt{\theta N} b_D \mu}{\sqrt{m}}$, then by Lemma E.2, we can bound the Rademacher complexity

$$\mathcal{R}_\mathcal{G}(\mathcal{F}_\gamma) \leq \inf_{\alpha>0}\left(\frac{4\alpha\gamma}{\sqrt{N}} + \frac{12}{N}\int_{\gamma\alpha}^{\gamma\sqrt{N}} \sqrt{\frac{v_1}{\epsilon^2} + v_2 \ln\frac{v_3}{\epsilon}}\, d\epsilon\right)$$

$$\overset{(a)}{\leq} \inf_{\alpha>0}\left(\frac{4\alpha\gamma}{\sqrt{N}} + \frac{12}{N}\int_{\gamma\alpha}^{\gamma\sqrt{N}} \sqrt{\frac{v_1 + v_2}{\epsilon^2} + v_2 \ln v_3}\, d\epsilon\right)$$

$$\overset{(b)}{\leq} \inf_{\alpha>0}\left(\frac{4\alpha\gamma}{\sqrt{N}} + \frac{12}{N}\left(\int_{\gamma\alpha}^{\gamma\sqrt{N}} \frac{\sqrt{v_1 + v_2}}{\epsilon} + \sqrt{v_2 \ln v_3}\, d\epsilon\right)\right)$$

$$= \inf_{\alpha>0}\left(\frac{4\alpha\gamma}{\sqrt{N}} + \frac{12}{N}\left(\sqrt{v_1 + v_2}\ln\left(\frac{\sqrt{N}}{\alpha}\right) + \gamma\sqrt{v_2 \ln v_3}\left(\sqrt{N} - \alpha\right)\right)\right)$$

$$\overset{(c)}{\leq} \frac{4\gamma}{N} + \frac{12\sqrt{v_1 + v_2}\ln N}{N} + \frac{12\gamma(N-1)\sqrt{v_2 \ln v_3}}{N\sqrt{N}}$$

$$\leq \frac{4\gamma}{N} + \frac{12\sqrt{v_1 + v_2}\ln N}{N} + \frac{12\gamma\sqrt{v_2 \ln v_3}}{\sqrt{N}}$$

where (a) holds due to $\ln \frac{1}{x} \leq \frac{1}{x^2}$, (b) holds due to $\sqrt{x+y} \leq \sqrt{x} + \sqrt{y}$. For (c), we have chosen $\alpha = \frac{1}{\sqrt{N}}$.

It can be shown that the covering number bound of $\mathcal{F}_D$ satisfies $\mathcal{N}(\epsilon, \mathcal{F}_D, \rho) \leq (3b_D/\epsilon)^{Kmd}$ (Lemma F.14). Combining the bounds of $\mathcal{N}(\epsilon, \mathcal{F}_G, \rho)$ and $\mathcal{N}(\epsilon, \mathcal{F}_D, \rho)$ and Lemmas E.1 and E.2, we derive the generalization bound:

$$L_\gamma(f) \leq \hat{L}_\gamma(f) + \frac{8\gamma + 24\sqrt{v_1 + v_2}\ln N + 24\gamma\sqrt{N v_2 \ln v_3}}{N} + 3\gamma\sqrt{\frac{\ln(2/\delta)}{2N}}$$

This finished the proof. $\qquad\square$

### F.7 Corollary of Theorem 3.2

**Corollary F.16** (Mis-classification rate upper bound of GRDL). *Let $n$ be the minimum number of nodes for graphs $\{G_i\}_{i=1}^N$, $\theta$ be the hyper-parameter in gaussian kernel (Equation (15)), $c = \|\tilde{\mathbf{A}}\|_\sigma$. For graphs $\mathcal{G} = \{(G_i, y_i)\}_{i=1}^N$ drawn i.i.d from any probability distribution over $\mathcal{X} \times \{1, \ldots, K\}$ and references $\{\mathbf{D}_k\}_{k=1}^K$, $\mathbf{D}_k \in \mathbb{R}^{m \times d}$, with probability at least $1 - \delta$, every margin $\zeta > 0$ and network $f \in \mathcal{F}$ under Assumption 3.1 satisfy*

$$\Pr_{G \sim \mathcal{X}}[\arg\max_j f(G)_j \neq y] \leq \hat{L}_\zeta(f) + 3\sqrt{\frac{\ln(2/\delta)}{2N}} + \frac{8 + 24\sqrt{v_1 + v_2}\ln N + 24\sqrt{N v_2 \ln v_3}}{N}$$

*where*

$$v_1 = \frac{256\theta K R_G}{n\zeta^2}, v_2 = Km\bar{d}, v_3 = \frac{48b_D\sqrt{\theta N}}{\sqrt{m}\zeta}, R_G = c^{2L}\|\mathbf{X}\|_2^2 \ln(2\bar{d}^2)\left(\prod_{l=1}^L \left(\prod_{i=1}^r \kappa_i^{(l)}\right)^2\right)\left(\sum_{l=1}^L \sum_{i=1}^r \left(\frac{b_i^{(l)}}{\kappa_i^{(l)}}\right)^{2/3}\right)^3,$$

*and $\hat{L}_\zeta(f) \leq N^{-1}\sum_i \mathbb{1}[f(G_i)_{y_i} \leq \zeta + \arg\max_{j \neq y_i} f(G_i)_j]$.*

*Proof.* Choose the loss $l_\gamma(\cdot, y)$ as

$$l_\gamma(\cdot, y) = r_\zeta(-\mathcal{M}(\cdot, y))$$

where $\mathcal{M}(v, y) := v_y - \max_{i \neq y} v_i$ is the margin operator and

$$r_\zeta(t) := \begin{cases} 0 & t < -\zeta, \\ 1 + t/\zeta & t \in [-\zeta, 0], \\ 1 & t > 0. \end{cases}$$

is called the ramp loss. The population ramp risk is defined as $L_\zeta(f) := \mathbb{E}_{G \sim \mathcal{X}}[r_\zeta(-\mathcal{M}(f(G), y)))]$. Given the graph dataset $\mathcal{G}$ sampled from $\mathcal{X}$, the empirical ramp risk is $\hat{L}_\zeta(f) := N^{-1}\sum_{i=1}^N r_\zeta(-\mathcal{M}(f(G_i), y_i))$. It is clear that $\mathbb{1}[\arg\max_j f(G)_j \neq y] \leq r_\zeta(-\mathcal{M}(f(G), y))$, so

$$\Pr_{G \sim \mathcal{X}}[\arg\max_j f(G)_j \neq y] = \mathbb{E}_{G \sim \mathcal{X}}\left[\mathbb{1}(\arg\max_j f(G)_j \neq y)\right] \leq L_\zeta(f)$$

It is easy to see that $\gamma = 1$ in this case. Also by Lemma F.6, $\mu = \frac{2}{\zeta}$. Then it is trivial to get the bound by Theorem 3.2 with simple substitution

$$\Pr_{G \sim \mathcal{X}}[\arg\max_j f(G)_j \neq y] \leq L_\zeta(f)$$

$$\leq \hat{L}_\zeta(f) + 3\sqrt{\frac{\ln(2/\delta)}{2N}} + \frac{8 + 24\sqrt{v_1 + v_2}\ln N + 24\sqrt{N v_2 \ln v_3}}{N}$$

where $v_1 = \frac{256\theta K R_G}{n\zeta^2}, v_2 = Km\bar{d}, v_3 = \frac{48b_D\sqrt{\theta N}}{\sqrt{m}\zeta}, R_G = $

$c^{2L}\|\mathbf{X}\|_2^2 \ln(2\bar{d}^2)\left(\prod_{l=1}^L \left(\prod_{i=1}^r \kappa_i^{(l)}\right)^2\right)\left(\sum_{l=1}^L \sum_{i=1}^r \left(\frac{b_i^{(l)}}{\kappa_i^{(l)}}\right)^{2/3}\right)^3.$ Also,

$r_\zeta(-\mathcal{M}(f(G_i), y_i)) \leq \mathbb{1}[f(G_i)_{y_i} \leq \zeta + \arg\max_{j \neq y_i} f(G_i)_j]$, so we have

$$\hat{L}_\zeta(f) = N^{-1}\sum_{i=1}^N r_\zeta(-\mathcal{M}(f(G_i), y_i)) \leq N^{-1}\sum_{i=1}^N \mathbb{1}[f(G_i)_{y_i} \leq \zeta + \arg\max_{j \neq y_i} f(G_i)_j]$$

$\square$

**Corollary F.17** (Generalization bound with cross-entropy loss). *Suppose $l_\gamma(\cdot, y)$ is the cross-entropy loss $\mathcal{L}_{CE}$ (7). Let $n$ be the minimum number of nodes for graphs $\{G_i\}_{i=1}^N$, $\theta$ be the hyper-parameter in gaussian kernel (Equation (15)), $c = \|\tilde{\mathbf{A}}\|_\sigma$. For graphs $\mathcal{G} = \{(G_i, y_i)\}_{i=1}^N$ drawn i.i.d from*

*any probability distribution over $\mathcal{X} \times \{1, \ldots, K\}$ and references $\{\mathbf{D}_k\}_{k=1}^K, \mathbf{D}_k \in \mathbb{R}^{m \times d}$, with probability at least $1 - \delta$, every network $f \in \mathcal{F}$ under Assumption 3.1 satisfy*

$$L_\gamma(f) \le \hat{L}_\gamma(f) + 3\gamma\sqrt{\frac{\ln(2/\delta)}{2N}} + \frac{8\gamma + 24\sqrt{v_1 + v_2}\ln N + 24\gamma\sqrt{Nv_2 \ln v_3}}{N}$$

*where* $v_1 = \frac{128\theta K R_G}{n}, v_2 = Km\bar{d}, v_3 = \frac{24\sqrt{2\theta N}b_D}{\sqrt{m}}, R_G =$

$c^{2L}\|\mathbf{X}\|_2^2 \ln(2\bar{d}^2)\left(\prod_{l=1}^L\left(\prod_{i=1}^r \kappa_i^{(l)}\right)^2\right)\left(\sum_{l=1}^L\sum_{i=1}^r\left(\frac{b_i^{(l)}}{\kappa_i^{(l)}}\right)^{2/3}\right)^3.$

*Proof.* According to Lemma F.7, $\mu = \sqrt{2}$. Then the proof is trivial by substituting it into Theorem 3.2. $\square$

### F.8 Adjacency matrix spectral norm

**Lemma F.18.** *Let $G = (V, E)$ be an undirected graph with adjacency matrix $\mathbf{A} \in \mathbb{R}^{n \times n}$, $d_G$ be the maximum degree of G. Then, the adjacency matrix satisfies*

$$\|\mathbf{A}\|_\sigma \le d_G$$

*Proof.* Based on the definition of the spectral norm, we have

$$\|\mathbf{A}\|_\sigma \overset{(a)}{=} \max_{\|\mathbf{x}\|_2=1} \mathbf{x}^\top \mathbf{A}\mathbf{x} = \max_{\|\mathbf{x}\|_2=1} \sum_{(i,j)\in E} x_i x_j \le \max_{\|\mathbf{x}\|_2=1} \sum_{(i,j)\in E} \frac{1}{2}(x_i^2 + x_j^2) = d_G \sum_{i\in V} x_i^2 = d_G$$

where (a) is because $\mathbf{A}$ is a real symmetric matrix. $\square$

**Lemma F.19.** *For any matrix $\mathbf{X} \in \mathbb{R}^{m \times n}$, $\|\mathbf{X}\|_\sigma \le \|\mathbf{X}\|_2 \le \sqrt{r}\|\mathbf{X}\|_\sigma$ where $r = rank(\mathbf{X})$*

*Proof.* By the definition of spectral norm,

$$\|\mathbf{X}\|_\sigma = \sqrt{\lambda_{\max}(\mathbf{X}^\top \mathbf{X})}$$

where $\lambda_{\max}$ denotes the largest eigenvalue. Since $\mathbf{X}^\top \mathbf{X}$ is a positive semi-definite real symmetric matrix, it must has $n$ real eigenvalues that can be ordered as

$$\lambda_1 \ge \lambda_2 \ge \cdots \ge \lambda_r > \lambda_{r+1} \cdots = \lambda_n = 0.$$

Then it has

$$\|\mathbf{X}\|_\sigma = \sqrt{\lambda_1} \le \sqrt{\sum_{i=1}^n \lambda_i} = \sqrt{\text{tr}(\mathbf{X}^\top \mathbf{X})} = \|\mathbf{X}\|_2 \le \sqrt{r\lambda_1} = \sqrt{r}\|\mathbf{X}\|_\sigma.$$

$\square$

**Lemma F.20.** *Let $G = (V, E)$ be a graph with adjacency matrix $\mathbf{A} \in \mathbb{R}^{n \times n}$. Assume $|E| > 0$, then $c = \|\tilde{\mathbf{A}}\|_\sigma = \|\mathbf{A} + \mathbf{I}\|_\sigma > 1$.*

*Proof.* By Lemma F.19,

$$\|\tilde{\mathbf{A}}\|_\sigma \ge \frac{1}{\sqrt{r}}\|\tilde{\mathbf{A}}\|_2 = \frac{1}{\sqrt{r}}\|\mathbf{A} + \mathbf{I}\|_2 \ge \frac{1}{\sqrt{r}}\|\mathbf{A}\|_2 + \frac{1}{\sqrt{r}}\|\mathbf{I}\|_2 \overset{(a)}{>} \frac{1}{\sqrt{r}}\|\mathbf{I}\|_2 = \sqrt{\frac{n}{r}} \ge 1$$

where (a) is because $|E| > 1$. $\square$

### F.9 Generalization of MMD (to be used in Section F.10)

Before stating the lemma, we first give alternative definitions of MMD. Let $\mathbb{P}$ be a continuous probability distribution of some random variable $Z$ taking values from space $\mathcal{Z}$. Then, the kernel mean embedding of $\mathbb{P}$ associated with the continuous, bounded, and positive-definite kernel function $k : \mathcal{Z} \times \mathcal{Z} \to \mathbb{R}$ is

$$\mu_{\mathbb{P}} := \int_{\mathcal{Z}} k(z, \cdot)\, d\mathbb{P}(z) \tag{35}$$

which is an element in the Reproducing Kernel Hilbert Space (RKHS) $\mathscr{H}$ associated with kernel $k$. In many practical situations, it is unrealistic to assume access to the true distribution $\mathbb{P}$. Instead, we only have access to samples $P = \{z_i\}_{i=1}^n$ from $\mathbb{P}$. We can approximate (35) by the empirical kernel mean embedding

$$\hat{\mu}_{\mathbb{P}} := \frac{1}{n} \sum_{i=1}^{n} k(z_i, \cdot). \tag{36}$$

For another continuous distribution $\mathbb{Q}$ with samples $Q = \{z_i'\}_{i=1}^m$, the MMD between the two probability distribution is defined as

$$\mathrm{MMD}(\mathbb{P}, \mathbb{Q}) = \|\mu_{\mathbb{P}} - \mu_{\mathbb{Q}}\|_{\mathscr{H}},$$

and the empirical MMD is

$$\mathrm{MMD}(P, Q) = \|\hat{\mu}_{\mathbb{P}} - \hat{\mu}_{\mathbb{Q}}\|_{\mathscr{H}}.$$

Denote $d := \mathrm{MMD}(\mathbb{P}, \mathbb{Q})$ and $\hat{d} := \mathrm{MMD}(P, Q)$, we have the follow Lemma:

**Lemma F.21.** *With probability at least $1 - \delta$ we have*

$$|d - \hat{d}| \leq \left(\frac{1}{\sqrt{m}} + \frac{1}{\sqrt{n}}\right)\left(2 + \sqrt{2 \log \frac{2}{\delta}}\right)$$

*Proof.* We use the following Lemma:

**Lemma F.22** (Theorem 7 of Gretton et al. [2012a], reformulated). *Assume $0 \leq k(x, y) \leq K$. Then with probability at least $1 - 2\exp\left(\frac{-\varepsilon^2 mn}{2K(m+n)}\right)$*

$$\left|d - \hat{d}\right| \leq 2\left(\frac{1}{\sqrt{m}} + \frac{1}{\sqrt{n}}\right) + \varepsilon$$

Let $\delta = 2\exp\left(\frac{-\varepsilon^2 mn}{2K(m+n)}\right)$, then it has

$$\varepsilon = \sqrt{\frac{1}{m} + \frac{1}{n}}\sqrt{2 \log\left(\frac{2}{\delta}\right)}.$$

Therefore, with probability at least $1 - \delta$

$$\left|d - \hat{d}\right| \leq 2\left(\frac{1}{\sqrt{m}} + \frac{1}{\sqrt{n}}\right) + \sqrt{\frac{1}{m} + \frac{1}{n}}\sqrt{2 \log\left(\frac{2}{\delta}\right)}$$

$$\leq \left(\frac{1}{\sqrt{m}} + \frac{1}{\sqrt{n}}\right)\left(2 + \sqrt{2 \log \frac{2}{\delta}}\right)$$

$\square$

## F.10 Proof of Lemma F.2

*Proof.* For simplicity, let $k' := \arg\min_{j \neq k} \text{MMD}(\mathbf{H}_i, \mathbf{D}_j)$. Since $\mathbf{H}_i$ and $\mathbf{D}_k$ are finite samples from $\mathbb{P}_k$, and $\mathbf{D}_{k'}$ is sampled from $\mathbb{P}_{k'}$, by Lemma F.21, we have

$$|\text{MMD}(\mathbb{P}_k, \mathbb{P}_k) - \text{MMD}(\mathbf{H}_i, \mathbf{D}_k)| \leq \Delta_1 = \left(\frac{1}{\sqrt{m}} + \frac{1}{\sqrt{n}}\right)\left(2 + \sqrt{2\log\frac{2}{\delta'}}\right) \quad w.p. \quad (1 - \delta'),$$

$$|\text{MMD}(\mathbb{P}_k, \mathbb{P}_{k'}) - \text{MMD}(\mathbf{H}_i, \mathbf{D}_{k'})| \leq \Delta_2 = \left(\frac{1}{\sqrt{m}} + \frac{1}{\sqrt{n}}\right)\left(2 + \sqrt{2\log\frac{2}{\delta'}}\right) \quad w.p. \quad (1 - \delta').$$

Therefore, with probability at least $1 - 2\delta'$ (union bound), we have

$$\text{MMD}(\mathbf{H}_i, \mathbf{D}_k) - \text{MMD}(\mathbb{P}_k, \mathbb{P}_k) = \text{MMD}(\mathbf{H}_i, \mathbf{D}_k) - 0 \leq \Delta_1 \quad \text{and} \quad \text{MMD}(\mathbb{P}_k, \mathbb{P}_{k'}) - \text{MMD}(\mathbf{H}_i, \mathbf{D}_{k'}) \leq \Delta_2.$$

It follows that

$$\text{MMD}(\mathbf{H}_i, \mathbf{D}_k) - \text{MMD}(\mathbf{H}_i, \mathbf{D}_{k'}) \leq -\text{MMD}(\mathbb{P}_k, \mathbb{P}_{k'}) + \Delta_1 + \Delta_2.$$

By Definition F.1, to ensure correct classification, we can let

$$-\text{MMD}(\mathbb{P}_k, \mathbb{P}_{k'}) + \Delta_1 + \Delta_2 < 0.$$

This means

$$\text{MMD}(\mathbb{P}_k, \mathbb{P}_{k'}) > \Delta_1 + \Delta_2 = \left(\frac{1}{\sqrt{m}} + \frac{1}{\sqrt{n}}\right)\left(4 + 2\sqrt{2\log\frac{2}{\delta'}}\right). \tag{37}$$

Therefore, if (37) holds, the classification is correct with probability at least $1 - 2\delta'$. Letting $\delta = 2\delta'$, we finish the proof. $\qquad\square$

## F.11 Proof of Theorem A.1

Let $\mathbf{H} \in \mathbb{R}^{\sum_i n_i \times d}$ be the output of the message passing layers. Then the mean readout is a matrix multiplication

$$\mathbf{Z} = \mathbf{MH}, \quad \mathbf{M} = \begin{bmatrix} \frac{1}{n_1} & \cdots & \frac{1}{n_1} & 0 & \cdots & 0 & 0 & \cdots & 0 \\ 0 & \cdots & 0 & \frac{1}{n_2} & \cdots & \frac{1}{n_2} & 0 & \cdots & 0 \\ \vdots & & \vdots & \vdots & & \vdots & \vdots & & \vdots \\ 0 & \cdots & 0 & 0 & \cdots & 0 & \frac{1}{n_N} & \cdots & \frac{1}{n_N} \end{bmatrix}.$$

It is easy to see $\|\mathbf{M}\|_\sigma \leq 1$. Since a MLP is concatenated after readout, by the proof of Lemma F.8, the covering number of GIN $\mathcal{F}'$ is

$$\ln\mathcal{N}(\epsilon, \mathcal{F}', \rho) \leq \frac{c^{2L}\|\mathbf{X}\|_2^2 \ln(2\bar{d}^2)}{\epsilon^2}\mathcal{A}$$

where $\mathcal{A} = \left(\prod_{l=1}^{L}(\prod_{i=1}^{r}\kappa_i^{(l)})^2(\prod_{i=1}^{r'}\kappa_i^{(L+1)})^2\right)\left(\sum_{l=1}^{L}\sum_{i=1}^{r}\left(\frac{b_i^{(l)}}{\kappa_i^{(l)}}\right)^{2/3} + \sum_{i=1}^{r'}\left(\frac{b_i^{(L+1)}}{\kappa_i^{(L+1)}}\right)^{2/3}\right)^3$.

Then by Lemma E.2, the generalization bound can be easily derived by taking $\alpha = \frac{1}{\sqrt{N}}$.

