# OpenReview forum: "Graph Classification via Reference Distribution Learning: Theory and Practice"
_NeurIPS.cc/2024/Conference — NeurIPS 2024 poster_

### Official Review · Reviewer_anD1 · 2024-06-21

**Soundness:** 2
**Presentation:** 3
**Contribution:** 2
**Rating:** 4
**Confidence:** 5

**Summary:**

Instead of compressing node embedding matrix into a graph-level vector, this paper proposes a reference distribution learning method GRDL especially designed for graph classification task. GRDL achieves graph classification by measuring the dissimilarity between distributions of the input graph and the references. Theoretical analysis of the generalization error bound offers guidance of hyperparameter tuning.

**Strengths:**

- This paper is well-motivated to avoid information loss caused by node embedding downsampling.
- The idea of considering graph classification as a distribution comparison problem is interesting.
- The authors have solid mathematical skills.

**Weaknesses:**

- The idea is borrowed from domain transfer learning but lacks adaption to graph classification task. In graph classification task, graphs in each dataset “are drawn i.i.d” as the authors mentioned at line 205. This means that although graphs have different classes, they belong to an identical distribution, which is contradictory to the hypothesis of this paper that different classes belong to different reference distributions.
- In my opinion, this paper didn’t avoid information compression compared to mean pooling and max pooling. MMD unifies the source and target inputs as a vertex in the same RKHS and then calculate the distance between them. I think the mapping from embedding matrix into a vertex can be regarded as information compression, and the calculation of mean discrepancy is similar to mean pooling.
- The time complexity is too high, the datasets used for large-scale experiments are unreasonable, and the competitors used in efficiency experiment is unreasonable. The time complexity is $O(N^2)$, which is the same as the highly criticised time-consuming node clustering pooling methods (such as DiffPool[1]), while node dropping methods (such as SAGP[2]) only requires a time complexity of $O(E)$. The complexity restricts its scalability to larger-scale graphs. Three so-called large-scale datasets are not really large-scale. They are only large in the number of graphs but not in the number of nodes in each single graph. The authors should add experiments on synthetic large-scale datasets. Besides, in time cost comparison, the authors didn’t choose competitors with SOTA efficiency but used two time consuming methods, which is quite unreasonable.
- The theoretical contribution of generalization error bound is limited. Firstly, the result offers fuzzy insights into the choose of hyperparameters by giving “moderate-size message passing GIN”, and “moderate-size references”. How to define “moderate”? At line 240, the authors analysis that “a network with a smaller $L4 and $r4 may guarantee a tighter bound on the population risk compared to a larger one. Therefore, a promising strategy is to use a moderate-size message passing GNN”. What’s the causality between “smaller” and “moderate-size”? That’s quite confusing. Experimental results in Appendix show that hyperparameters are still choosed on trial and error. Besides, the generalization ability comparison with GIN with mean readout is meaningless because the competitor is not the SOTA.
- Important baselines are missing in the experiments. Please add comparison results with recent graph pooling methods such as ASAP[3], MinCutPool[4], StructPool[5], MuchPool[6], TAP[7], Wit-TopoPool[8], and MSGNN[9].

References

[1]Ying Z, You J, Morris C, et al. Hierarchical graph representation learning with differentiable pooling[J]. Advances in neural information processing systems, 2018, 31.

[2]Lee J, Lee I, Kang J. Self-attention graph pooling[C]//International conference on machine learning. PMLR, 2019: 3734-3743.

[3]Ranjan E, Sanyal S, Talukdar P. Asap: Adaptive structure aware pooling for learning hierarchical graph representations[C]//Proceedings of the AAAI conference on artificial intelligence. 2020, 34(04): 5470-5477.

[4]Bianchi F M, Grattarola D, Alippi C. Spectral clustering with graph neural networks for graph pooling[C]//International conference on machine learning. PMLR, 2020: 874-883.

[5]Yuan H, Ji S. Structpool: Structured graph pooling via conditional random fields[C]//Proceedings of the 8th International Conference on Learning Representations. 2020.

[6]Du J, Wang S, Miao H, et al. Multi-Channel Pooling Graph Neural Networks[C]//IJCAI. 2021: 1442-1448.

[7]Gao H, Liu Y, Ji S. Topology-aware graph pooling networks[J]. IEEE Transactions on Pattern Analysis and Machine Intelligence, 2021, 43(12): 4512-4518.

[8]Chen Y, Gel Y R. Topological pooling on graphs[C]//Proceedings of the AAAI Conference on Artificial Intelligence. 2023, 37(6): 7096-7103.

[9]Lv Y, Tian Z, Xie Z, et al. Multi-scale Graph Pooling Approach with Adaptive Key Subgraph for Graph Representations[C]//Proceedings of the 32nd ACM International Conference on Information and Knowledge Management. 2023: 1736-1745.

**Questions:**

Please refer to the weaknesses.

---

> ### Author Rebuttal · Authors · 2024-08-04
>
> **Response to Weakness 1:**
>
> Thanks for your comment. This is a misunderstanding. Almost all papers studying the theory of classification use the i.i.d assumption. The identical distribution does **NOT** contradict the multiple classes scenario. Here the identical data distribution $\mathcal{D}$ is actually a composition of $K$ sub-distributions, where each sub-distribution (related to one reference distribution in our method) corresponds to a class. An intuitive example is the Gaussian mixture models: data distribution $\mathcal{D}$ is composed of $K$ Gaussians, i.e., $p_{\mathcal{D}}\left(\mathbf{x}\right)=\sum_{k=1}^K p\left(\mathbf{x} \mid \mathbf{z}\right) p(\mathbf{z})=\sum_{k=1}^K \pi_k \mathcal{N}\left(\mathbf{x} \mid \mu_k, \Sigma_k\right)$ (samples drawn i.i.d from $p_{\mathcal{D}}$ belongs to different classes). For instance, the following popular papers on classification used i.i.d assumption as us.
>
> [1] VN Vapnik. The nature of statistical learning theory. Springer science \& business media, 2013. (68000+ citations)
>
> [2] PL Bartlett et al. Spectrally-normalized margin bounds for neural networks. NeurIPS 2017. (1200+ citations)
>
> [3] C. Zhang et al. Understanding deep learning (still) requires rethinking generalization. Communications of the ACM 64, no. 3 (2021): 107-115. (7000+ citations)
>
> **Response to Weakness 2:**
>
> We treat the nodes' embeddings as discrete distributions and measure the distance between two distributions using MMD. MMD [1] with a Gaussian kernel actually compares all orders of statistics between two distributions rather than the means, though this is implicitly conducted in the RKHS using the mean difference. In other words, MMD compares also the higher-order statistics between distributions while mean pooling compares only the first-order statistics, i.e., mean. Therefore, the information loss of MMD is much less than the mean and max poolings.
>
> We used Example 2.1 to show the limitation of mean and max pooling and the motivation for we regard each node embedding matrix as a discrete distribution. Two different distributions may have the same mean and max but their MMD is never zero. MMD has also been utilized in generative models (e.g. MMD-GAN [2]) to compare distributions, where comparing the means (like mean pooling) in the feature space given by a neural network does not work.
>
>
> [1] Arthur Gretton et al. A kernel two-sample test. JMLR 2012.
>
> [2] Li et al. MMD GAN: Towards deeper understanding of moment matching network. NeurIPS 2017.
>
> **Response to Weakness 3:**
>
> We acknowledge that our method is efficient in terms of the number of graphs rather than the number of nodes in each graph. However, there are many graph datasets with a large number of graphs but a relatively small number of nodes in each graph, for which our method can be used.
>
> It is really very difficult to propose an algorithm with both SOTA accuracy and SOTA efficiency. Our method has a good trade-off between them. We follow your suggestion and compare the time cost of our method with two pooling methods published in 2023 (WitTopoPool and MSGNN). The results are in the **global rebuttal** due to the character limitation. In fact, these two methods are much more costly than ours. The GNN-based topological pooling layer requires the calculation of pairwise node similarity, which is $O(N^2)$. Besides, the time complexity of its witness complex-based topological layer is also quadratic. The subgraph sampling, selection, and evolution of MSGNN are also very costly.
>
> **Response to Weakness 4:**
>
> Thanks for your insightful comment. Similar to the previous work on learning theory such as [1][2][3], our theorem can only show the impact of the model architecture and parameters on the generalization ability of the model and there is always a trade-off between model complexity and training accuracy, that's why we used the words like "moderate" and "smaller". The upper bounds of the training error are data-dependent, which means we may never point out exactly what model is the best according to only the theoretical analysis.
>
> The generalization bound is the upper bound of the difference between training error and testing error. We say "a network with a smaller $L$ and $r$ may guarantee a tighter bound on..." because in the bound, the term $\tilde{\mathcal{O}}\left(\frac{\mu \bar{b}\|\mathbf{X}\|_2 c^L(L r)^{\frac{3}{2}} \bar{\kappa}^{L r} \sqrt{\theta K / n}}{N}+\gamma \sqrt{\frac{K m d}{N}}\right)$ increases with $L$ and $r$. So here the "smaller" is an exact expression.
>
> We say "moderate-size references" because the bound scales the reference size $m$ as $\tilde{O}(\sqrt{m})$, which means the bound is not very sensitive to $m$ because of the square root. If $m$ is too small, the expressive power of the model will be low; if it is very large, the complexity $\tilde{O}(\sqrt{m})$ is high. Therefore, we say "moderate-size", meaning a trade-off, should be used. This is supported by the experiments in Appendix D.3 (Figure 6) in our paper.
>
> Let's use an intuitive example (may not be true in practice) to further explain why we have to use "moderate" rather than a concrete value (e.g. 0, 1, $\infty$) in a discussion of the theoretical result. Suppose there is a parameter or hyperparameter $s$ of the model and the bounds scales with it as $\tilde{O}((s-1)^2)$ and $s$ does not influence the training error, then the best $s$ is 1. However, in practice, the training error may decrease when $s$ increases. The training error is data-dependent, which means we cannot find the best $s$ using the theoretical result only. Instead, one may use cross-validation or AutoML to find a good $s$.
>
> **Response to Weakness 5:**
>
> The added results (MinCutPool, ASAP, Wit-TopoPool, MSGNN) are in **global rebuttal**. Our method has better classification accuracy on most of the datasets and has the highest average accuracy.
>
> **Please do not hesitate to let us know if you need more explanation or have further questions.**

---

> ### Author Response · Authors · 2024-08-13
>
> Dear Reviewer anD1,
>
> We appreciate your comments and suggestions. Did our response address your concerns? We are keen to receive your feedback and provide further explanation if necessary.
>
> Sincerely,
>
> Authors

---

### Official Review · Reviewer_aUnT · 2024-07-11

**Soundness:** 4
**Presentation:** 3
**Contribution:** 4
**Rating:** 7
**Confidence:** 5

**Summary:**

The paper introduces a novel algorithm called GRDL for graph classification. GRDL  treats each graph’s latent node embeddings given by GNN layers as a discrete distribution and directly classify distributions without global pooling. The authors derived generalization error bounds for GRDL and verified them numerically. The experiments on many graph datasets show the superiority of GRDL. GRDL is 10 times faster than leading competitors.

**Strengths:**

* Originality: The proposed algorithm GRDL and the theoretical results are novel.
* Quality: The paper is well-organized and contains rich theoretical results and numerical comparisons.
* Clarity: The motivation (e.g. Example 2.1), assumptions, optimization, implication of theorems, and the experimental setting have been clearly explained.
* Significance: Graph classification is a challenging problem due to the difficulty in converting nodes’ embeddings to a vector as the global representation of each graph. The proposed method gets rid of readout operations and directly classifies the discrete distributions formed by nodes’ embeddings. It can outperform graph kernels, GIN, and graph transformers. Moreover, the paper proved that the proposed model has better generalization ability than the baseline GIN, which is a big contribution.

**Weaknesses:**

There is no major weakness found. Please refer to my questions in the next section.

**Questions:**

* Besides GIN, there are other GNN models such as GCN and GAT. Why did the authors consider GIN only in the theoretical analysis and experiments?
* I think the parameter $\theta$ of the Gaussian kernel can be absorbed into the neural network parameter. I suggest the authors discuss the necessity of setting or optimizing $\theta$ separately.
* In Theorem 3.2, the bound is related to $K$, the number of classes. But, in many literature of generalization analysis, the bound is not related to the number of classes. Could the author explain the difference?

**Limitations:**

The authors discussed the limitations in Section 6.

---

> ### Author Rebuttal · Authors · 2024-08-04
>
> **Response to Question 1:**
>
> Thank you for your question. We chose GIN because it is a highly representative model within the class of Graph Neural Networks (GNNs) that utilize neighbor aggregation schemes. GIN is probably the most expressive model in this category, and such schemes are widely employed in many classic GNN designs, including GCN and GraphSAGE.
>
> In terms of theoretical analysis, our approach can be readily adapted to models like GCN and GraphSAGE with only minor modifications. GAT uses an attention mechanism for aggregation, and this approach differs significantly from the general aggregation scheme employed by GIN, GCN, and many other models. We believe the analysis of GIN provides broader and more valuable insights. Additionally, as we said in our paper, the analysis on GIN's generalization ability is currently limited, and our work aims to address this gap. It's also worth noting that our proposed reference layer can be combined with other models like GCN and GAT to classify graphs without any modification.
>
> **Response to Question 2:**
>
> Thank you for your insightful comment. While it is theoretically possible to absorb the parameter $\theta$ of the Gaussian kernel into the neural network parameters, we believe that setting or optimizing $\theta$ separately is crucial in practice.
>
> The main reason for this is that the initial neural network parameters and the reference distributions do not necessarily ensure that nodes of graphs $\mathbf{x}$ are close to nodes of graphs $\mathbf{x}'$. If $\theta$ or the scale of the node embeddings $\mathbf{H}$ is too large, the Gaussian kernel becomes overly sharp, resulting in almost zero values. This can lead to a situation where the Maximum Mean Discrepancy (MMD) fails to effectively quantify the distance between the embeddings and the reference distributions, of which the complexity is related to $K$.
>
> To illustrate the importance of setting or optimizing $\theta$ separately, we actually conducted experiments on the MUTAG dataset with different values of $\theta$ on Appendix D6, Table 11. The results, shown in the following table, demonstrate that the choice of $\theta$ significantly impacts classification accuracy.
>     $$
>     \begin{matrix}
>     \hline
>     \mathbf{\theta} & 1\times10^{-4} & 1\times10^{-3} & 1\times10^{-2} & 1\times10^{-1} & 1 & 1\times10^{1} & 1\times10^{2} & 1\times10^{3}\\\\
>     \hline
>     \textbf{Accuracy} & 0.9096 & 0.9149 & 0.9113 & 0.9113 & 0.8254 & 0.6822 & 0.5737 & 0.3345\\\\
>     \hline
>     \end{matrix}
>     $$
> This shows the necessity of carefully selecting or optimizing $\theta$ to ensure good classification performance. We will elaborate on these findings in the revised manuscript.
>
> **Response to Question 3:**
>
> Thank you for your insightful question. The difference arises from our model's unique use of reference distributions. In previous literature on generalization analysis, the focus has often been on models that aggregate node embeddings into vectors and then classify these vectors using another neural network, without incorporating reference distributions. This conventional approach does not typically involve a dependency on the number of classes $K$. In contrast, our model introduces reference distributions.
>
> We have also included an additional generalization analysis theorem in the appendix (Theorem A.1) for the GIN model with mean pooling, which does not involve reference distributions. This analysis, consistent with traditional approaches, does not include $K$ in the bound. We hope this clarification and the additional theorem help highlight the unique aspects of our model and the corresponding theoretical analysis.
>
> **Please do not hesitate to let us know if you need more explanation or have further questions.**

---

> > ### Comment · Reviewer_aUnT · 2024-08-09
> >
> > The rebuttal and the additional experimental results have adequately addressed my concerns. Therefore, I raise my confidence level to 5.

---

> > > ### Author Response · Authors · 2024-08-09
> > >
> > > We sincerely appreciate your recognition of our work.

---

### Official Review · Reviewer_5HH3 · 2024-07-13

**Soundness:** 2
**Presentation:** 3
**Contribution:** 3
**Rating:** 6
**Confidence:** 3

**Summary:**

This paper proposes to make graph-level predictions via distribution comparison between node-level representations and discrete reference distributions. The authors claim that their proposed method avoids the requirements of graph pooling for graph-level tasks. and reduce the risk of information loss. Theoretical and empirical justification results are provided.

**Strengths:**

- The authors propose a novel and simple method for graph classification and provide theoretical analysis on the generalization bound. The discussion is clear and extensive.
- The method shows advantage in time cost compared to related works.
- The authors performed extensive ablation study on the proposed method.

**Weaknesses:**

- Equation in L138-139 (a missing equation index?) also requires summation over nodes. The description of "avoid graph pooling operation" seems to be overclaimed. I suggest that the authors reconsider it.
- An ablation study on the discrimination loss and the usage of node-level representation is required. I think the main improvement in performance may be attributed to the discrimination loss. It helps the model learn distant representations for graphs of different labels in the feature space.  The performance of the proposed method should be compared with and without the discrimination loss. Besides, the authors should also implement a baseline with discrimination loss where node representations are first sum-pooling and then compared with the reference distribution.
- Minor: Latest baseline models are required for a comprehensive empirical comparison, such as graph pooling including SEP[1], GMT[2], and CoCN[3] and graph transformers including Exphormer[4], GRIT[5], and MPNN-VN[6]. Considering the time limitation and the similarity of the datasets, a comparison on part of the datasets will be sufficient.

[1] Structural entropy guided graph hierarchical pooling. ICML'22

[2] Accurate Learning of Graph Representations with Graph Multiset Pooling. ICLR'22

[3] All in a row: Compressed convolution networks for graphs. ICML'23

[4] Exphormer: Sparse Transformers for Graphs. ICML'23

[5] Graph Inductive Biases in Transformers without Message Passing. ICML'23

[6] On the Connection Between MPNN and Graph Transformer. ICML'23

**Questions:**

Please refer to the weaknesses.

**Limitations:**

Yes. The authors describe the limited performances of the proposed method on certain datasets.

---

> ### Author Rebuttal · Authors · 2024-08-04
>
> **Response to Weakness 1:**
>
> Thank you for your feedback. The equation in lines 138-139 stems from the definition of the Maximum Mean Discrepancy (MMD). MMD is a measure between two discrete distributions, where the inputs are two matrices and the output is a scalar. This means that in the computation, elementary operations like summation are compulsory: it involves summation over the kernel values rather than the original node features. Regarding the claim about avoiding graph pooling operations, we would like to clarify that traditional graph pooling operations typically compress a graph's node embedding matrix into a single vector before classification. In contrast, our method allows for the direct classification of the graph using its node embeddings without compressing them into a single vector. We treat the node embedding matrix as a discrete distribution and handle it by MMD, a theoretical-grounded measure between two distributions. We appreciate your suggestion and will ensure that this distinction is more clearly communicated in the revised manuscript.
>
> **Response to Weakness 2:**
>
> Thank you for your insightful suggestion. We appreciate your interest in the impact of the discrimination loss on our model's performance. We actually included an ablation study of our proposed method without the discrimination loss in Appendix D.5 Table 10 (where the discrimination loss coefficient $\lambda = 0$). Following your second advice, we also implement a baseline (denoted as GRDL:SumDis) with discrimination loss, where node representations are first aggregated using sum-pooling and then compared with the reference distribution. The results of the ablation study regarding these two baselines are in the following table.
>     $$
>     \begin{matrix}
>     \hline
>     Method              & MUTAG         & PROTEINS      & NCI1          & IMDB-B        & IMDB-M        & PTC-MR        & BZR           & COLLAB & Average \\\\
>     \hline
>     GRDL        & \textbf{92.1} \pm 5.9 & \textbf{82.6} \pm 1.2 & \textbf{80.4}\pm 0.8          & \textbf{74.8}\pm 2.0 & \textbf{52.9} \pm 1.8          & \textbf{68.3} \pm 5.4          & \textbf{92.0} \pm 1.1 & \textbf{79.8}\pm 0.9          & \textbf{77.9} \\\\
>     GRDL:\lambda = 0   & 89.9 \pm 4.9  & 81.8 \pm 1.3  & 80.0\pm 1.6   & 73.1 \pm 1.5  & 51.3 \pm 1.4  & 66.6 \pm 5.9  & 89.5 \pm 2.3  & 79.0 \pm 1.0 & 76.4\\\\
>     GRDL: SumDis & 89.9 \pm 6.0 & 78.4 \pm 0.6 & 77.2 \pm 1.7 & 71.6 \pm 5.2 & 49.8 \pm 5.4 & 62.5 \pm 6.3 & 85.3 \pm 1.5 &  77.1 \pm 0.9 & 74.0\\\\
>     \hline
>     \end{matrix}
>     $$
>
> The results show that discrimination loss can increase the performance of our model. However, our method without discrimination loss still outperforms the baseline with sum-pooling and discrimination loss. We will include these results in the revised version of the manuscript.
>
>
> **Response to Weakness 2 (minor):**
>
> Thank you so much for pointing out these baselines. Due to the time limit, we implemented SEP, GMT, and MPNN-VN. The results are shown in the following table. Our method GRDL outperformed the competitors in most cases. We will include these additional results in our paper and discuss all the six references you mentioned.
>     $$
>     \begin{matrix}
>     \hline
>     \text{Method}  & MUTAG                 & PROTEINS              & NCI1                  & IMDB-B                & IMDB-M                & PTC-MR                & BZR                   & COLLAB                       & Average \\\\
>     \hline
>     \text{GRDL (ours)}   & \textbf{92.1} \pm 5.9 & \textbf{82.6} \pm 1.2 & 80.4 \pm 0.8 & \textbf{74.8} \pm 2.0 & \textbf{52.9} \pm 1.8 & 68.3 \pm 5.4 & \textbf{92.0} \pm 1.1 & 79.8\pm 0.9 & \textbf{77.9} \\\\
>     \text{SEP  [ICML 2022]}   & 89.4 \pm 6.1  & 76.4 \pm 0.4  & 78.4 \pm 0.6   & 74.1 \pm 0.6  & 51.5 \pm 0.7  & 68.5 \pm 5.2  & 86.9 \pm 0.8  & \textbf{81.3} \pm 0.2 & 75.8\\\\
>     \text{GMT [ICLR2022]}    & 89.9 \pm 4.2  & 75.1 \pm 0.6  & 79.9 \pm 0.4   & 73.5 \pm 0.8  & 50.7 \pm 0.8  & 70.2 \pm 6.2  & 85.6 \pm 0.8  & 80.7 \pm 0.5 & 75.7\\\\
>     \text{MPNN-VN [ICML2023]} & \textbf{92.1} \pm 5.2  & 78.3 \pm 1.0  & \textbf{80.9} \pm 0.8   & 72.4 \pm 1.2  & 50.9 \pm 1.9  & \textbf{71.4} \pm 5.2  & 90.2 \pm 1.1  & 80.1 \pm 0.8 & 77.0\\\\
>     \hline
>     \end{matrix}
>     $$
>
> **Please do not hesitate to let us know if you need more explanation or have further questions.**

---

> ### Comment · Reviewer_5HH3 · 2024-08-12
>
> I appreciate the detailed response which has addressed my concerns. The new ablation results (W2) further validate the proposed method and the selected comparison results seem promising. Please make sure to clarify the difference between the pre-pooling methods and your discrepancy measuring and summation strategy, and update your manuscript based on the rebuttal.

---

> > ### Author Response · Authors · 2024-08-12
> >
> > Thank you so much for your feedback and support. The suggestions from you and the other two reviewers have helped us improve the quality of our paper. We will update the manuscript according to the rebuttal.

---

### Author Rebuttal · Authors · 2024-08-05

We sincerely appreciate the reviewers' comments. In this rebuttal, we added the following experiments.
1. **Training time per epoch** compared to two latest baselines (WitTopoPool and MSGNN both proposed in 2023) mentioned by reviewer anD1 on both experimental datasets in our paper and three synthetic datasets. The three synthetic datasets have 2000 graphs with 100(SYN-100), 300(SYN-300), 500(SYN-500), and 1000(SYN-1000) nodes per graph, respectively. The edge number is $0.1n^2$ where $n$ is the number of nodes. See the following table.
$$
\begin{matrix}
\hline
  & MUTAG                 & PROTEINS              & NCI1                  & IMDB-B                & IMDB-M                & PTC-MR                & BZR    & COLLAB & SYN-100 & SYN-300 & SYN-500 & SYN-1000\\\\
\hline
\text{GRDL (ours)} & 0.4 & 3.4 & 12.6 & 2.4 & 3.5 & 0.8 & 1.2 & 16.3 & 26.6 & 45.8 & 88.7 & 220.8\\\\
\text{WitTopoPool (2023)}  & 0.4  & 2.6 & 21.4 & 2.4 & 2.6 & 1.0 & 1.3 & 39.1 & 32.9 & 50.8 & 97.5 & 201.3\\\\
\text{MSGNN (2023)} & 45.2  & - & - & - & - & 75.5  & 135.3 & - & - & - & - & -\\\\
\hline
\end{matrix}
$$
Message to Reviewer anD1: As can be seen, our method GRDL is more efficient than these two latest pooling methods you mentioned when the number of nodes in each graph is less than 1000. In fact, graph dataset for graph-level learning (not node-level learning) with an average node number larger than 1000 is rare.
2. **Ablation study** for the two terms in the objective function of our method GRDL. See the following table.
    $$
    \begin{matrix}
    \hline
    Method              & MUTAG         & PROTEINS      & NCI1          & IMDB-B        & IMDB-M        & PTC-MR        & BZR           & COLLAB & Average \\\\
    \hline
    GRDL        & \textbf{92.1} \pm 5.9 & \textbf{82.6} \pm 1.2 & \textbf{80.4}\pm 0.8          & \textbf{74.8}\pm 2.0 & \textbf{52.9} \pm 1.8          & \textbf{68.3} \pm 5.4          & \textbf{92.0} \pm 1.1 & \textbf{79.8}\pm 0.9          & \textbf{77.9} \\\\
    GRDL:\lambda = 0   & 89.9 \pm 4.9  & 81.8 \pm 1.3  & 80.0\pm 1.6   & 73.1 \pm 1.5  & 51.3 \pm 1.4  & 66.6 \pm 5.9  & 89.5 \pm 2.3  & 79.0 \pm 1.0 & 76.4\\\\
    GRDL: SumDis & 89.9 \pm 6.0 & 78.4 \pm 0.6 & 77.2 \pm 1.7 & 71.6 \pm 5.2 & 49.8 \pm 5.4 & 62.5 \pm 6.3 & 85.3 \pm 1.5 &  77.1 \pm 0.9 & 74.0\\\\
    \hline
    \end{matrix}
    $$
 3. **In addition to the 12 baselines** compared in our original submission, we **added 6 more baselines**. See the following table.
 $$
    \begin{matrix}
    \hline
    Method  & MUTAG                 & PROTEINS              & NCI1                  & IMDB-B                & IMDB-M                & PTC-MR                & BZR                   & COLLAB                       & Average \\\\
    \hline
    GRDL (ours)    & \textbf{92.1} \pm 5.9 & \textbf{82.6} \pm 1.2 & \textbf{80.4} \pm 0.8 & \textbf{74.8} \pm 2.0 & \textbf{52.9} \pm 1.8 & 68.3 \pm 5.4 & \textbf{92.0} \pm 1.1 & 79.8\pm 0.9 & \textbf{77.9} \\\\
    SEP (2022)    & 89.4 \pm 6.1  & 76.4 \pm 0.4  & 78.4 \pm 0.6   & 74.1 \pm 0.6  & 51.5 \pm 0.7  & 68.5 \pm 5.2  & 86.9 \pm 0.8  & \textbf{81.3} \pm 0.2 & 75.8\\\\
    GMT (2022)    & 89.9 \pm 4.2  & 75.1 \pm 0.6  & 79.9 \pm 0.4   & 73.5 \pm 0.8  & 50.7 \pm 0.8  & \textbf{70.2} \pm 6.2  & 85.6 \pm 0.8  & 80.7 \pm 0.5 & 75.7\\\\
    MinCutPool (2020)  & 90.6 \pm 4.6  & 74.7 \pm 0.5  & 74.3 \pm 0.9   & 72.7 \pm 0.8  & 51.0 \pm 0.7  & 68.3 \pm 4.4  & 87.2 \pm 1.0  & 80.9 \pm 0.3 & 75.0\\\\
    ASAP (2020)  & 87.4 \pm 5.7  & 73.9 \pm 0.6  & 71.5 \pm 0.4   & 72.8 \pm 0.5  & 50.8 \pm 0.8  & 64.6 \pm 6.8  & 85.3 \pm 1.3  & 78.6 \pm 0.5 & 73.1\\\\
    WitTopoPool (2023) & 89.4 \pm 5.4  & 80.0 \pm 3.2  & 79.9 \pm 1.3   & 72.6 \pm 1.8  & 52.9 \pm 0.8  & 64.6 \pm 6.8  & 87.8 \pm 2.4  & 80.1 \pm 1.6 & 75.9\\\\
    MSGNN (2023) & 78.4 \pm 7.1  & - & - & - & - & 56.4 \pm 6.6  & 78.1 \pm 2.5 & - & \\\\
    \hline
    \end{matrix}
    $$
Message to Reviewer anD1: Our method outperformed the four pooling methods you mentioned as well as the other two pooling methods SEP and GMT proposed in 2022.

Besides supplementary experiments, we made several important rebuttals:
* We explained why our method avoids pooling: MMD takes summation over kernel values instead of summation over original nodes (see responses to reviewer 5HH3).
* We added an ablation study on the discrimination loss (see responses to reviewer 5HH3).
* We added experiments of six additional baselines and compared the classification accuracy (see responses to reviewer 5HH3, anD1 and also the PDF file in global rebuttal).
* We explained why optimizing/setting $\theta$ separately is necessary (see responses to reviewer aUnT).
* We argued the use of i.i.d assumption in our theoretical analysis is correct and has been commonly used in many famous papers of learning theory (see responses to reviewer anD1).
* We added experiments of two SOTA methods' training time per epoch on both datasets used in our paper and three synthetic datasets (see responses to reviewer anD1 and also the PDF file in global rebuttal).

**We are looking forward to your feedback on our rebuttal. Thanks.**

---

> ### Comment · Reviewer_5HH3 · 2024-08-08
>
> The authors referred to a PDF file in their global rebuttal, but I am unable to locate it. Could the authors please confirm whether this file has been correctly uploaded?

---

> > ### Author Response · Authors · 2024-08-08
> >
> > Sorry for the confusion. All additional experiments can be found in the global rebuttal. We did not update the PDF. Thanks for pointing it out.

---

### Decision · Program_Chairs · 2024-09-25

**Decision:**

Accept (poster)

**Comment:**

The paper proposes an architecture for graph-level prediction based on node-level representation computing by a GNN. Instead of using a pooling method, the paper derives a reference distribution learning method for graph-learning tasks. That is, the authors' architecture measures the dissimilarity between distributions of the input graph and the references. The authors propose a generalization analysis based on standard tools and conducted an empirical study showing somewhat promising results, mostly on smaller datasets from the TUDataset collection.

The reviewers liked the novel idea of using distribution comparison for graph-level prediction. Some highlighted the clear motivation and good exposition and the performed extensive ablation studies on the proposed method. Reviewer anD1 argued that the method shares similarities with mean pooling and max pooling; the time complexity is too high, and some GNN pooling baselines are missing in the experiments. The authors partially addressed some of these concerns in the rebuttal.

This is a borderline paper. A majority of reviewers favor its acceptance due to its novelty. I follow this assessment.